ecology, palaeontology

Caribbean, competition, extinction, global change, invertebrates, molluscs

**Author for correspondence:**
Christine D. Bacon
e-mail: christinedbacon@gmail.com

# Selective extinction against redundant species buffers functional diversity

Catalina Pimiento[1,2,†], Christine D. Bacon[3,4,†], Daniele Silvestro[3,4,5], Austin Hendy[6], Carlos Jaramillo[2,7,8], Alexander Zizka[4,9], Xavier Meyer[5,10] and Alexandre Antonelli[3,4,11]

[1]Department of Biosciences, Swansea University, Swansea SA2 8PP, UK
[2]Smithsonian Tropical Research Institute, Box 0843-03092, Balboa, Ancon, Republic of Panama
[3]Department of Biological and Environmental Sciences, University of Gothenburg, Box 461, SE-405 30 Gothenburg, Sweden
[4]Gothenburg Global Biodiversity Centre, Box 461, SE-405 30 Gothenburg, Sweden
[5]Department of Biology, University of Fribourg, Fribourg, Switzerland
[6]Natural History Museum of Los Angeles County, Los Angeles, CA 90007, USA
[7]Equipe de Paléontologie, Institut des Sciences de l'Évolution de Montpellier, University of Montpellier, Place Eugène Bataillon, 34095 Montpellier Cedex 05, France
[8]Institut des Sciences de l'Évolution de Montpellier, University of Montpellier, Place Eugène Bataillon, 34095 Montpellier Cedex 05, France
[9]German Center for Integrative Biodiversity Research (iDiv) Halle Jena Leipzig, 04103 Leipzig, Germany
[10]Department of Integrative Biology, University of California, Berkeley, CA 94720, USA
[11]Royal Botanical Gardens Kew, Richmond TW9 3AE, UK

CP, 0000-0002-5320-7246; CDB, 0000-0003-2341-2705

The extinction of species can destabilize ecological processes. A way to assess the ecological consequences of species loss is by examining changes in functional diversity. The preservation of functional diversity depends on the range of ecological roles performed by species, or functional richness, and the number of species per role, or functional redundancy. However, current knowledge is based on short timescales and an understanding of how functional diversity responds to long-term biodiversity dynamics has been limited by the availability of deep-time, trait-based data. Here, we compile an exceptional trait dataset of fossil molluscs from a 23-million-year interval in the Caribbean Sea (34 011 records, 4422 species) and develop a novel Bayesian model of multi-trait-dependent diversification to reconstruct mollusc (i) diversity dynamics, (ii) changes in functional diversity, and (iii) extinction selectivity over the last 23 Myr. Our results identify high diversification between 23–5 Mya, leading to increases in both functional richness and redundancy. Conversely, over the last three million years, a period of high extinction rates resulted in the loss of 49% of species but only 3% of functional richness. Extinction rates were significantly higher in small, functionally redundant species suggesting that competition mediated the response of species to environmental change. Taken together, our results identify long-term diversification and selective extinction against redundant species that allowed functional diversity to grow over time, ultimately buffering the ecological functions of biological communities against extinction.

## 1. Introduction

Traditionally, the effects of species extinctions have been assessed based on changes in the taxonomic structure of communities. However, extinctions can trigger ecological responses well beyond taxonomic counts. One way to assess the ecological impact of extinctions is by assessing changes in functional diversity [1]. Functional diversity identifies the range of ecological roles present in a community by linking taxonomic identity to species traits such as size, diet, locomotion, and habitat [2]. Previous studies on functional diversity changes

†Contributed equally to this work.

after extinction events have found that global marine invertebrate faunas, which are usually small-sized and species-rich, tend to be functionally resilient [3–6], while the communities of much larger and scarce marine vertebrates tend to be highly vulnerable [7,8]. It is widely recognized that the maintenance of functional diversity depends on the level of functional redundancy (i.e. the number of species performing broadly similar functions; [9,10]), which stabilizes ecosystem processes and allows ecosystem resilience [11]. However, the role of redundancy in conferring functional resilience after extinctions, as well as the functional responses to extinction over deep timescales, remains understudied.

Functional redundancy can be an indicator of competitive interactions [12]. For example, redundant species (i.e. those sharing functional traits, and therefore, similar ecologies) can face high competition for resources. In the presence of competition, increasing standing diversity can reduce speciation and/or increase extinction rates [13–15]. Models of such diversity-dependent diversification typically assume that all species within a clade [16] or among clades [17] compete identically for resources. In reality, however, the strength of competition between species can vary depending on whether or not they share similar ecologies and functions (i.e. functional traits). The integration of functional aspects of species across evolutionary timescales can therefore provide a better understanding of the effects of competition on extinctions.

Marine molluscs have a rich fossil record, which includes some of the most complete and best-preserved macrofaunal sequences of the Cenozoic [6,18]. A widely documented extinction event of benthic species in the Caribbean occurred in the Plio-Pleistocene [19–22]. Traditionally, the causes of this extinction event have been attributed to regional oceanographic changes associated with the closure of the Central American Seaway (hereafter, CAS) in the late Cenozoic, between 4 and 2 Ma [23], including increases in salinity and carbonate, and declines in both seasonality and benthic habitat [20,24–26].

In the particular case of Caribbean molluscs, it has been shown that, in agreement with the proposed environmental drivers, taxa associated with high planktonic productivity such as suspension-feeding bivalves and predatory gastropods were particularly affected by this extinction [20,21,23,27]. However, it has also been suggested that the extinction of molluscs presented a 2-million-year temporal lag between environmental change and detectable extinction, becoming extinct between only 2 and 1 Mya [24]. Despite the general agreement on the timing and causes of the extinction of Caribbean benthic molluscs, new geological and biological evidence suggests that the CAS constricted much earlier than previously proposed, during the late Miocene, around 10 Mya [28–34]. As such, questions on the mechanisms underlying this extinction event remain open.

Caribbean molluscs provide an ideal model to understand the ecological responses of communities to extinction over long timescales [3,5,6]. They have an unusually rich fossil record and have persisted through major environmental changes. Despite the widespread extinction event experienced by Caribbean faunas in the recent geological past, today the Caribbean Sea is a biogeographic zone categorized as a biodiversity hotspot [35,36]. Nevertheless, the entire region is vulnerable to anthropogenic threats such as acidification, climate change, invasive species, and overfishing [37].

Here, we studied a 23-million-year fossil sequence of Caribbean molluscs based on a novel dataset comprising 34 011 records representing 4422 species (electronic supplementary material, figure S1). We assessed the ecological response of the species assemblage to climate change and extinction by asking (i) what are the diversification dynamics of Caribbean molluscs? (ii) how did functional diversity change through time? and (iii) does extinction follow diversity-dependent dynamics? Our results show how marine assemblages responded to global change and extinction through geological time. Further, our study exemplifies the importance of functional redundancy in counteracting the effects of species loss and highlights the role of competition in mediating species response to climate change.

## 2. Results and discussion

### (a) Diversification dynamics

We assessed the diversification dynamics of Caribbean molluscs by quantifying speciation and extinction rates, net diversification, and species richness during the last 23 Myr. To do so, we applied a Bayesian approach that accounts for sampling and preservation biases and dating uncertainties (see Material and methods and electronic supplementary material) to fossil occurrences identified at the species level. We found eight (95% credible interval, CI: 4–15) significant rate shifts in speciation rates and seven significant shifts (95% CI: 4–12) in extinction rates, which are punctuated across the evolutionary history of Caribbean molluscs (electronic supplementary material, figure S2). After a gradual decrease during the early Miocene, speciation rates were generally constant during the middle and late Miocene. Then, there was an increase during the early Pliocene, peaking around the early Pliocene/late Pliocene boundary (approx. 3.7 Ma), and sharply decreasing thereafter (figure 1a). Extinction rates were roughly constant and lower than speciation during the Miocene (figure 1b). During the Pliocene and Pleistocene (from approximately 5 Ma onwards), there was a progressive increase in extinction rates, reaching maximum values around the Pliocene/Pleistocene boundary, approximately 2.6 Ma (figure 1b). Extinction rates remained consistently high throughout the Pleistocene (figure 1b). Net diversification rates were positive throughout the Miocene and until the early Pliocene (figure 1c), followed by a negative phase from the late Pliocene to the Pleistocene, between 3.6 and 2.9 Ma (grey band in figure 1). Finally, there was an almost continuous increase in species richness during the Neogene, with maximum diversity reached in the late Pliocene and resulting in a 49% net loss from the late Pliocene to the Pleistocene (figure 1d).

Our results indicate that molluscan communities experienced major diversity changes during the last 23 Myr, which were characterized by two phases: (i) an increase in diversity during most of the Neogene (between 23 and 5 Ma, peaking around 3.7 Ma) and (ii) an extinction phase at end of the Neogene and throughout the Quaternary (from 5 Ma onwards, peaking around 2.6 Ma). During the first phase, diversity gradually increased over an approximately 18-million-year period of constantly low speciation and extinction rates (figure 1a,b). This phase culminated in the early Pliocene when speciation rates peaked, allowing species richness to reach maximum values (figure 1d). The diversity

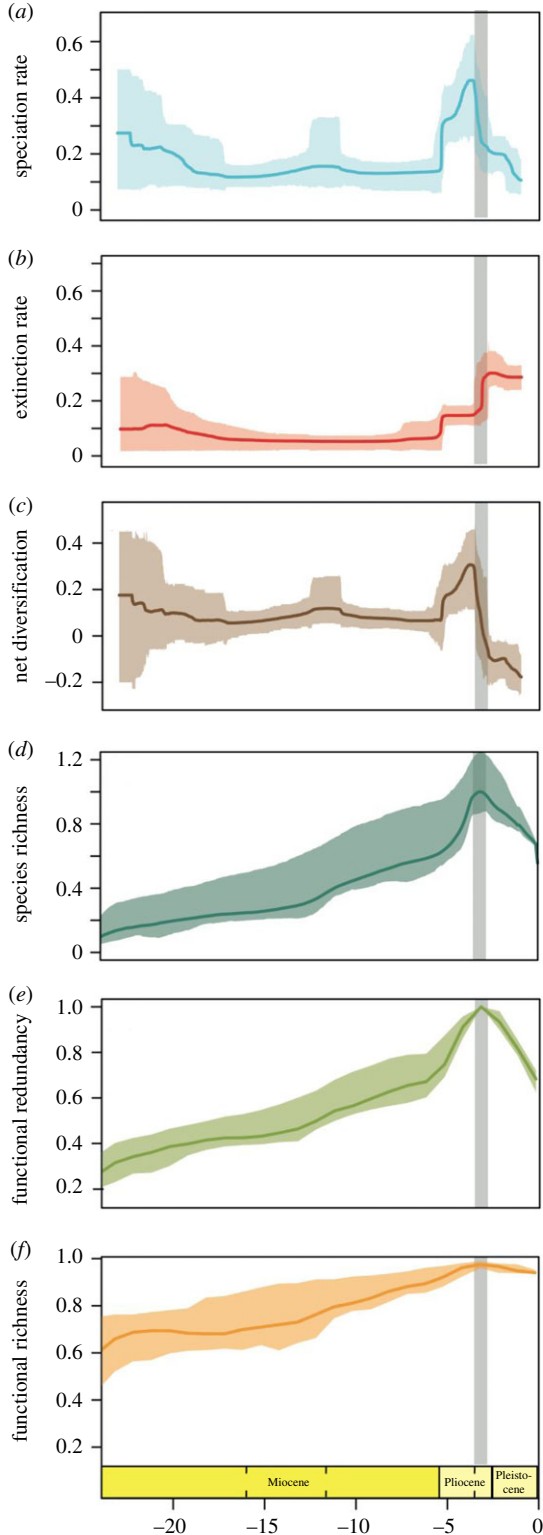

**Figure 1.** Diversity trajectories of Caribbean molluscs from the Neogene to the Quaternary. (*a*) Speciation rates. (*b*) Extinction rates. (*c*) Net diversification rates. (*d*) Species richness (proportion relative to maximum number of species). (*e*) Functional redundancy (mean number of species per functional entity (FE or unique trait combinations); proportion relative to maximum value). (*f*) Functional richness (proportional volume of functional space occupied). Box over the *x*-axis show geological epochs; dashed lines inside divide early, middle, late Miocene, and early and late Pliocene. Grey band denotes the time frame when the net diversification drops to a negative value for the first time due to increased extinction and decreased speciation (between 2.9 and 3.6 Ma). Polygons around the lines in a, b, and c represent credible intervals. Polygons in d, e, and f are confidence intervals derived from resampling the longevity of each species (see Methods). (Online version in colour.)

increase phase was coeval with the closing of the CAS [28–34], which gradually provided additional substrate, thereby increasing habitat complexity [38,39].

The phase of positive diversification was followed by a 5-million-year phase of high extinction (figure 1*b*) and negative net diversification (grey band; figure 1*c*), which resulted in a net loss of 49% of species. This phase roughly falls within the time range of elevated extinction rates previously reported during the formation of the Isthmus of Panama (figure 3d in [24]). However, our results contrast in terms of when the peak took place. While we found the extinction rates to be highest starting approximately 2.6 Ma, O'Dea *et al.* [24] found extinction to peak between 2 and 1 Ma. Contrasting results can be due to differences in geographical range covered (Panama versus Caribbean), taxonomic resolution (genus versus species), and analytical methods used to estimate extinction rates (number of last occurrences divided by the total number of taxa present per million years versus Bayesian inference that accounts for sampling and preservation biases; see Material and methods and electronic supplementary material). Of particular relevance is the difference in geographic range, as the use of the Caribbean area as a whole allowed us to sample greater habitat heterogeneity during the entire time series.

The time of the previously proposed extinction peak was interpreted as a lagged response to the environmental changes attributed to the closure of the CAS, between 2.25 and 3.45 Ma [24]. Based on our results and on the new evidence suggesting that the CAS constricted earlier, during the late Miocene (approx. 10 Ma) [28–34], we offer an alternative interpretation. We suggest that the high extinction rates of Caribbean molluscs at the end of the Neogene were likely a response to the environmental changes associated with the Northern Hemisphere glaciation, which started in the late Pliocene and persisted throughout the Pleistocene [40–43]. These climatic changes caused an abrupt transition to a new climatic state dominated by colder and more variable global temperatures [40–43]. Importantly, these climate changes produced large sea-level oscillations [40,42] and a significant reduction in the extent of sublittoral areas available [7], decreasing habitat for benthic communities in the Caribbean.

In sum, based on our results and recent evidence concerning the timing of the closure of the CAS, we interpret the positive diversification that took place during most of the Neogene as related to the additional substrate in the Caribbean Sea as the CAS was closing [38,39]. By contrast, the extinction of nearly half of Caribbean molluscs by the Quaternary was likely related to the unstable and reduced habitats that those species faced as a consequence of the Northern Hemisphere glaciation from the late Pliocene onwards [7,40,42].

## (b) Functional diversity

In order to understand the ecological effects of the diversification of Caribbean molluscs, we quantified changes in functional diversity through time. We did so by performing trait-based analyses following the methods described in Mouillot *et al.* [1] and as applied to the fossil record in Pimiento *et al.* [7]. Accordingly, we assigned five ecological traits that characterize molluscan functional diversity (i.e. body size, life habit, locomotion, environment, and diet; see electronic supplementary material, table S1 and methods) to (i) quantify the number of functional entities

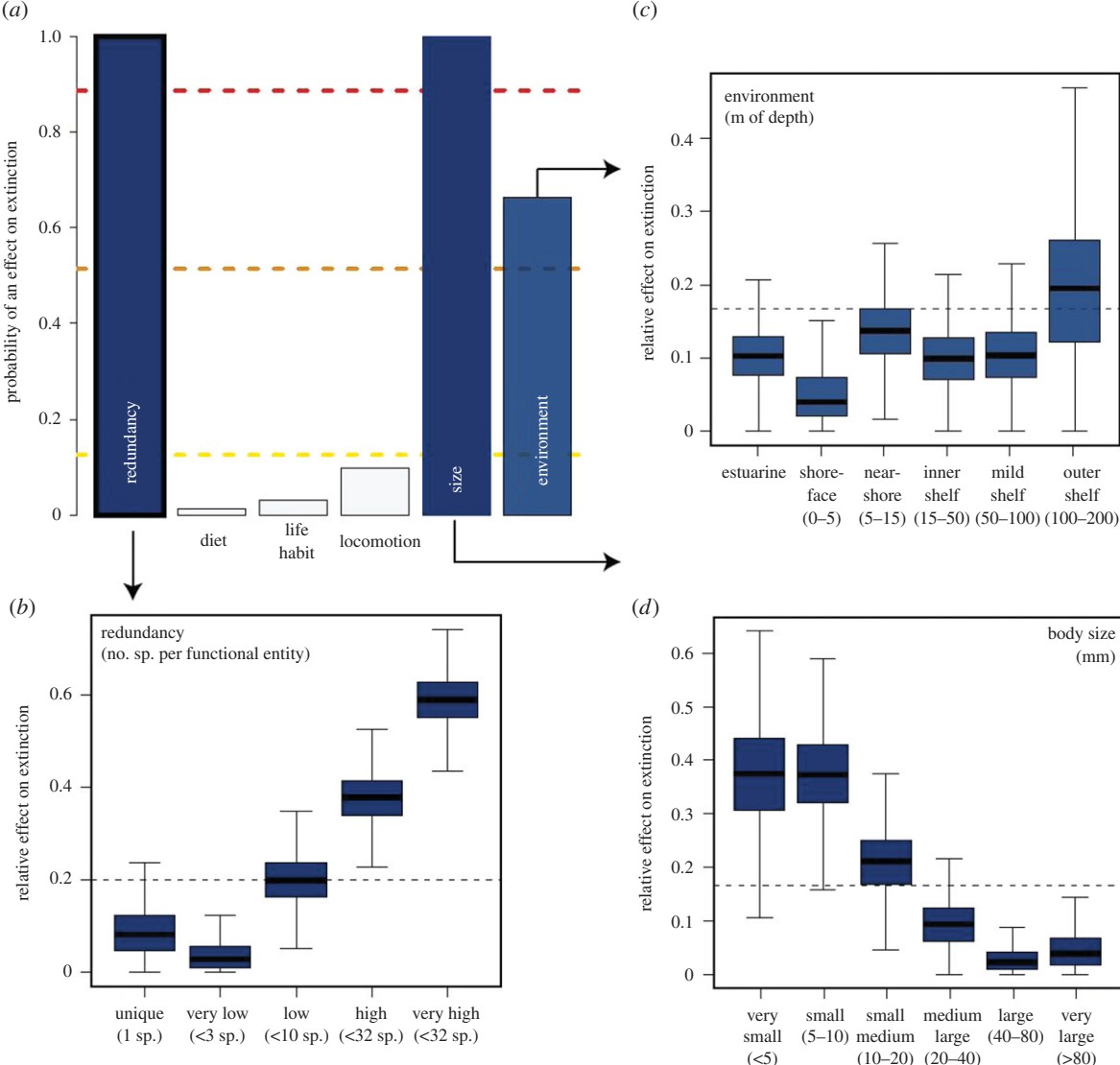

**Figure 2.** The relative effect of traits on the extinction of Caribbean molluscs. (*a*) Posterior probability of inclusion for each of the five traits plus redundancy in the multi-trait-dependent extinction analyses. Redundancy bar is different from the others to denote that it is not a trait. Dashed lines indicate the thresholds corresponding to log Bayes factor values of 2 (yellow, bottom line), 6 (orange, middle line), and 10 (red, top line) units represent positive, strong, and very strong statistical support. Blue shades indicate the strength of the statistical evidence, as ranked by Bayes factors, supporting an effect of the traits on extinction rates. (*b–d*) Effect of extinction for traits with the strongest statistical support. The *y*-axis corresponds to the Dirichlet-distributed *m* multipliers (see Methods, equation (3.1)). Dashed line is the expected value of the multipliers under the null model where the trait has no effect on extinction. Since all traits are analysed in a single model, the extinction rates of each species are a function of all six effects. Thus, the effects of the individual traits are interpreted as their impact on extinction, all else being equal. (Online version in colour.)

(i.e. unique trait combinations; hereafter FEs) and (ii) construct a trait space per time bin (see Material and methods). We found a general increase from the Miocene to the early Pliocene in the number of FEs (electronic supplementary material, figure S3; 58% increase); the mean number of species per FE, hereafter *functional redundancy* (figure 1*e* and electronic supplementary material, figure S4; 68% increase); the over-representation of some FEs (see Material and methods), hereafter *over-redundancy* (electronic supplementary material, figure S3; 18% increase); and in the convex hull volume of the trait space (see Material and methods), hereafter *functional richness* (figure 1*f*; 31% increase). All of these metrics peak between the early Pliocene and late Pliocene (between 4 and 3 Ma). While functional redundancy sharply decreases afterwards (by 31%), the number of FEs, over-redundancy, and functional richness remain virtually unchanged (9%, 3%, and 3.5% decrease, respectively; figure 2*e,f* and electronic supplementary material, figure S2). We further compared these

results against null expectations (see Material and methods) and found that all functional diversity values were no different than expected for most of the Neogene (electronic supplementary material, figure S3; electronic supplementary material, table S2). Conversely, from the late Miocene to the Quaternary, we found that: (i) functional redundancy values were higher than expected in the late Miocene, and mostly as expected thereafter; (ii) over-redundancy was mostly as expected, and (iii) the number of FEs and functional richness values were generally significantly higher than expected (electronic supplementary material, figure S3; and table S2).

Our analyses suggest that Caribbean molluscan assemblages experienced an overall increase in functional diversity during the Miocene and Pliocene. Such an increase of functional diversity was characterized by the appearance of new functions (as evidenced by the increased number of FEs), as opposed to the over-representation of pre-existing ones (as evidenced by a lower increase of functional

over-redundancy during the same time period; electronic supplementary material, figure S3). The general increase in functional diversity during most of the Neogene was coupled with the growth of species richness that took place during the same time (figure 1). Indeed, our simulations indicate functional diversity values were as expected given the number of species present in the assemblage (electronic supplementary material, figure S3; and table S2). Nevertheless, from the late Miocene onwards, when extinction rates started to increase and species richness started to decline, the number of FEs and functional richness were largely maintained and therefore, decoupled from species richness (figure 1f; electronic supplementary material, figure S3). As such, at the time of the highest extinction rates and consequent species loss, FEs and functional richness are significantly higher than expected (electronic supplementary material, table S2).

Our results indicate that the extinction phase of Caribbean molluscs, which started in the late Miocene (approx. 5 Ma) and peaked approximately 2.6 Ma, had a disproportionally small effect on functional diversity (figure 1f; electronic supplementary material, figure S3; and table S2). These results are consistent with previous studies documenting negligible changes in global functional diversity in the face of mass extinction events [3,4,6,44]. The combined evidence suggests that the functional resilience of benthic assemblages to extinction is consistent across spatial scales (regional to global) and extinction events (moderate to large, i.e. 'mass extinctions'). Further, the decoupling of species loss and functional diversity suggests that extinction was non-random.

## (c) Extinction selectivity

We tested if extinction followed diversity-dependent dynamics (i.e. if there was extinction selectivity) by developing a novel multi-trait-dependent extinction model that we introduce here (see Material and methods). Briefly, we tested the effect of functional redundancy (i.e. number of species per functional entity) and ecological traits (i.e. body size, life habit, locomotion, environment, and diet; electronic supplementary material, table S1) on overall extinction rates. We found strong evidence that extinction did not affect all species equally (figure 2). Functional redundancy and body size had the highest effect on extinction rates (strong statistical support; log Bayes factor greater than 10; red line in figure 2) followed by environment (i.e. water depth; positive statistical support; log Bayes factor greater than 6; orange line in figure 2a). Accordingly, very high redundancy (species sharing their FEs with more than 32 species; figure 2b; also see Material and methods) had the most significant effect on extinctions (their entire range of relative effect on extinction was higher than expected; figure 2b). High redundancy (species sharing their FEs with 10 to 32 species; figure 2b) also had an important, albeit relatively weaker effect on extinctions. Similarly, very small and small size (less than 10 mm; figure 2d) were associated with higher extinction rates. Finally, shore-face environments (less than 5 m of depth; figure 2c) were less affected by extinction than all other environments (figure 2c,d). None of the other traits—such as diet, life habitat, and locomotion—had a significant effect on extinction rates (figure 2a).

Our multi-trait dependence models indicate that redundancy and size had a much stronger effect on extinctions

than all other ecological traits, with small, highly redundant species being significantly more affected by extinctions than those that were large or functionally unique (figure 2). This result, combined with our finding that during the period of elevated extinction rates (from 5 Ma onwards; figure 1) Caribbean molluscs maintained significantly higher functional richness than expected by chance (electronic supplementary material, figure S3), suggests that selectivity against redundant species conferred functional resilience to extinction across assemblages. This is because the loss of redundant species likely allowed the spread of species among functional groups, preventing increases in over-redundancy, generally stabilizing the community, and ultimately enabling new FEs to persist. Indeed, extinction selectivity against rich FEs, or asymmetric functional selectivity, had been previously hypothesized as a possible explanation for the disproportional persistence of FEs, even during mass extinction events [6]. Similarly, the extinction susceptibility of highly redundant species suggests that competition may have played a major role in driving extinctions. This could be due to the fact that highly redundant species (i.e. those sharing traits and therefore, ecological functions) would have been competing more for resources than unique species, especially during times of large sea-level oscillations and habitat loss. Accordingly, we propose that biotic interactions could have mediated the response of species to the environmental changes that took place during times of high extinction rates.

We also found that body size had an important effect on extinctions (figure 2), with small-sized molluscs being more susceptible than medium- and large-sized species (figure 2d). The association between body size and extinction risk has been widely studied in the literature [e.g. 45–51]. In general, large size has been associated with historical extinctions and current risk (but see [46]), whereas large- and small-sized organisms have been equally at risk in the geological past [45]. In the fossil record of marine invertebrates, the effect of body size on extinctions has been found to vary across extinction events [45]. Our finding of selectivity against small-sized Caribbean molluscs reinforces our hypothesis of competitive interactions having an important role on extinction (figure 2) because smaller organisms tend to be more redundant [52,53] and therefore, face higher competition than larger organisms.

Consistent with our current knowledge on the habitat loss and sea-level oscillations that took place as Caribbean molluscs entered the extinction phase [7,20,40,42], our models found statistical support for an environmental effect on extinction rates (figure 2a). Within the neritic zone (i.e. above 200 m of depth, the zone in which all our samples lived; also see electronic supplementary material, dataset), species living in shore-face environments (less than 5 m of depth; also called intertidal; electronic supplementary material, table S1) had the lowest extinction risk. We suggest that these species were better equipped to tolerate the dramatic eustatic changes that took place from the late Pliocene onwards, as they would constantly face large sea-level changes during their lifetime [54] in comparison to species living in other environments (e.g. [55]). Although we found that environment, as defined here in terms of water depth, was an important factor in the extinction of Caribbean molluscs, our results show that redundancy and body size were the major determinants of extinction risk.

Diet, life style, and locomotion have been proposed as drivers of the extinction of molluscs at the end of the Neogene in

the Caribbean, with infaunal suspension and predatory feeders being more affected than attached epifaunal deposit feeders [20,21,23,27]. These results led previous authors to hypothesize that changes in benthic diversity at the time were caused by a decrease in productivity (as indicated by an increase of suspension feeder abundance) and an increase in the development of seagrass meadows and coral reefs (as indicated by an increase in abundance of attached deposit feeders) [20,21,23,27]. By contrast, we found no effect of diet, life style, or locomotion on extinction rates (figure 2). Nevertheless, our finding that species living in environments between 0 and 5 m of depth had the lowest extinction risk (figure 2c) is consistent with the hypothesis of seagrasses enabling the survival of Caribbean molluscs. It is important to note that although statistically significant, the effect of environment (i.e. water depth) on extinction rates in Caribbean molluscs during the last 23 Myr shows no consistent trend (figure 2c) and should therefore be interpreted with caution. We further suggest the contrasting results presented here relative to previous studies could be due to the fact that we used different traits and trait resolution [20,21,23,27].

### (d) Conclusions

Our study of Caribbean molluscs over the last 23 Ma reveals two distinct phases in terms of diversification and functional diversity: an approximately 18-million-year long phase of coupled taxonomic and functional diversity increase that resulted in maximum levels of functional redundancy; followed by an extinction phase that started in the late Miocene (approx. 5 Ma), peaked around 2.6 Ma, and resulted in negative net diversification (starting between 3.6 and 2.9 Ma) and the loss of 49% of species. Importantly, the extinction phase had negligible impacts on functional richness (figure 1). Further, our novel multi-trait dependence model identified selective pressure against small, functionally redundant species (figure 2). Based on our results we propose that (i) the response of species to environmental change at the end of the Neogene was determined by species competition within redundant FEs and that (ii) non-redundant species were generally released from competition, enabling the maintenance of functional richness in molluscan communities during times of extinction and thereby conferring ecological resilience.

## 3. Material and methods

### (a) Dataset

We compiled a dataset consisting of 34 011 fossil occurrences, representing 960 genera and 4422 species of Caribbean molluscs ranging from the Neogene to the Pleistocene/Holocene boundary (23.03–0.01 Ma; electronic supplementary material, dataset). The data was collected at the species level from the published literature and unpublished field observations available from the Paleobiology Database (https://paleobiodb.orgl) and museum-deposited species available from iDigBio (https://www.idig-bio.org; both downloaded 23/12/2018). While each data portal maintains a taxonomic backbone, the nomenclature of these independent datasets was standardized using the World Register of Marine Species taxon match service for extant species and genera. This eliminated many synonyms, misspelling, and consistently enforced taxonomic ranks. Extinct genera and species were accommodated within this taxonomic framework, with

care taken to eliminate spelling variations. We defined the study area of the Caribbean following the marine ecoregions of the world [56] (as available at www.worldwildlife.org). Accordingly, our dataset only includes fossils from the 'Tropical Northwest Atlantic' province (between 5.7° N and 35.7° S). To identify relevant records, we used the speciesgeocodeR package v. 1.8 in R [57]. In a few cases ([11]; figure 1), we assigned sites that are located outside any of the provinces and are terrestrial today, to the geographically closest province, using the 'dist2Line' function of the geosphere package v. 1.5–10 [58]. The age range of each occurrence was further refined using published references to place them into the finest stratigraphic resolution possible. To do so, we followed the latest geological timescale [59] and the estimate of 2.58 Ma for the Pliocene/Pleistocene boundary [60].

### (b) Speciation and extinction rates

We estimated speciation and extinction rates of Caribbean molluscs through a Bayesian analysis of all Neogene and Quaternary fossil occurrences identified to species level, while accounting for incomplete sampling due to fossilization and preservation processes and dating uncertainties. The stratigraphic ranges assigned to each occurrence were treated as dating uncertainty [61]. We resampled the age of each occurrence randomly from a uniform distribution spanning its minimum and maximum age and repeated this procedure 10 times to generate 10 resampled datasets. All analyses described below were repeated on the 10 replicates and their results combined in a joint posterior sample to incorporate dating uncertainty in the results [61]. Three main sets of parameters were simultaneously estimated using PyRate v. 2 [61,62]: (i) preservation rates quantifying the expected number of fossil occurrences per sampled lineage per time unit (1 Myr), (i) speciation and extinction times for each species, which usually extend beyond first and last fossil occurrences, and (iii) speciation and extinction rates for each species (expected number of speciation and extinction events per lineage per Myr) and their temporal variation. Details on how preservation was modelled and on how rate shifts were calculated can be found in the electronic supplementary material.

### (c) Functional diversity analyses

We selected five traits (i.e. body size, life habit, locomotion, environment, and diet; electronic supplementary material, table S1) that characterize the ecological roles of Caribbean molluscs [63–65]. Traits were assigned to species using authoritative taxon-specific texts, online databases, and expert assessments based on both extant relatives and the fossil record (electronic supplementary material, dataset). There were limited differences in the number of species occurrences per trait category (electronic supplementary material, figure S6). Based on the trait assignments, we calculated a series of functional indices in order to evaluate changes in functional composition and structure across the Neogene and Quaternary (23.03–0.01 Ma). To make this comparison, we used the longevity of each species estimated in PyRate (i.e. the time of origination and extinction of each species; see above) to record the presence and absence of species in 23 time bins of 1 Myr each (from 23–0 Ma). Accordingly, all functional diversity analyses described below were made per time bin. Given that the age of each species at a given time bin was assigned based on their longevity, no sampling differences between time bins were detected. Nevertheless, in order to incorporate age uncertainties, we resampled the estimated longevity of each species 100 times and performed 100 iterations of all functional diversity calculations. For each time bin, we evaluated: (i) the functional composition of Caribbean molluscs based on FEs or unique trait combinations (which reflects the diversity of ecological roles) and (ii) their functional structure

based on their functional space (which reflects the range of species' functional strategies).

### (i) Functional entities, redundancy, and over-redundancy

We used the R functions 'species_to_FE' and 'FE_metrics' [66] to calculate the number of FEs, the level functional redundancy (i.e. the mean number of species per functional entity [5,10], and the level of over-redundancy (i.e. the percentage of species that fill FEs above the mean level of functional redundancy, that is, the over-representation of some FEs) per time bin. We identified 7776 FEs for Caribbean molluscs, which correspond to the total possible combinations of traits and states. However, the species in our sample only occupy 4.53% of those (352 FEs in total). Throughout the Neogene and the Quaternary, there are, on average, 204 FEs per time bin (modal value = 279; electronic supplementary material, table S3) and five species per functional entity (modal value = six species per functional entity; electronic supplementary material, table S3). The functional redundancy values over time were robust to the number of traits used (see electronic supplementary material, figure S7).

### (ii) Functional richness

We created a multidimensional functional space by applying the methods of Mouillot et al. [1]. To do so, we first created a species trait distance matrix using the 'gowdis' function of the FD package [67] to retrieve the axes of a Principal Coordinate Analysis (PCoA; using the function 'dbFD'). Using the 'quality_funct_space' R function [68] we determined that our data is best represented using four dimensions, or PCoA axes. Based on this space, and using the 'multidimFD' function [66], we then calculated functional richness as the percentage of the convex hull volume occupied in each time bin relative to the overall functional space.

### (d) Multi-trait-dependent extinction model

To assess whether extinction rates varied among lineages as a function of one or multiple discrete traits, we developed a novel multi-trait-dependent extinction (MTE) model, which was included in the PyRate software. We express the likelihood of the estimated longevity of species $i$, here indicated with $t_i$, based on the birth-death model described by Silvestro et al. [61] and assuming a lineage-specific extinction rate $\mu_i$. Thus, the likelihood of an extinct lineage $i$ is computed as

$$P(t_i|\mu_i) \propto \mu_i \times \exp(-\mu_i t_i),$$

where $t_i$ is the lineage duration until extinction, while the likelihood of an extant lineage $j$ is computed as

$$P(t_j|\mu_j) \propto \exp(-\mu_j t_j),$$

where $t_j$ is the lineage duration until the present, or until the most recent boundary of the time window of interest.

In the MTE model, we parameterized the extinction rate as a function of a mean extinction rate ($\mu_0$) and Dirichlet-distributed multipliers $m$. The multipliers express the trait-dependent component of the model, and quantify the deviation of extinction from the mean associated with the lineage-specific trait state $v$. The resulting rate for a lineage $i$ in a set of $S$ species is

$$\mu_i = \mu_0 \times \frac{m_{v_i}}{\sum_j^S m_{v_j}} \times S, \text{ where } v \in \{0, \ldots, N\} \text{ and } m_v \sim \text{Dir}(\alpha_1, \ldots, \alpha_N).$$

(3.1)

Under this parameterization, $\mu_0$ equals the mean extinction rate across all lineages in a given time window, i.e.

$$\mu_0 = \frac{\sum_j^S \mu_j}{S}.$$

Here, we extended the model to allow for multiple traits to have an effect on extinction rates, by making the lineage-specific extinction rates a function of the combined trait states:

$$\mu_i = \mu_0 \times \frac{\prod_v^V m_{v_i}}{\sum_j^S (\prod_v^V m_{v_j})} \times S,$$

where $V$ is the number of traits included in the model.

We implemented the MTE model in a Bayesian framework to estimate all the free parameters, namely the mean extinction rate ($\mu_0$) and the Dirichlet-distributed multipliers ($m$). We used the Markov chain Monte Carlo (MCMC) algorithm to obtain posterior samples of all parameters.

We used an exponential prior on the mean extinction rate with rate parameter equal to 1. Efficient sampling of Dirichlet-distributed variables is typically quite problematic; therefore, we used here a re-parameterization to (i) improve MCMC convergence and (ii) facilitate the use of hyper-priors on the shape parameters of the Dirichlet distribution ($\alpha_1, \ldots, \alpha_N$). We followed the parameterization proposed by Gelman et al. [69] and expressed the trait-dependent multipliers as

$$m_i = \frac{\exp(y_i)}{\sum_j^N \exp(y_j)}.$$

(3.2)

We fixed $y_1 = 0$ (as only $N-1$ values are necessary to define $N$ Dirichlet-distributed values) and used normal prior distributions on the $y$ parameters with equal means (i.e. the prior expectation is that extinction rates are constant across lineages and independent of the trait) and with precision parameter $\tau$:

$$y_i \sim N(0, \tau).$$

(3.3)

When multiple traits are analysed ($V > 1$) each trait is assigned an independent precision parameter $\tau$. The precision parameter(s) $\tau$ were assigned a gamma hyper-prior:

$$\tau \sim \Gamma(\alpha = 1.5, \beta),$$

(3.4)

with fixed shape parameter $\alpha = 1.5$, and rate $\beta$. The rate $\beta$ was itself considered as a free parameter and assigned a vague exponential hyper-prior:

$$\beta \sim \Gamma(\alpha_0 = 1, \beta_0 = 0.1).$$

(3.5)

This parameterization improves substantially the efficiency of the posterior sampling because it allows us to sample several (hyper)parameters directly from their conjugate posterior distribution. For a given set of $y$ values and $\beta$, we sample the precision parameters $\tau$ directly from a gamma conjugate distribution:

$$P(\tau|\mathbf{y}, \alpha, \beta) \sim \Gamma\left(\alpha + \frac{N}{2}, \beta + \frac{1}{2}\sum_{i=1}^N (y_i)\right).$$

Likewise, posterior values of the rate parameter $\beta$ of the gamma prior on $\tau$ are sampled directly from its gamma distribution for a given set of precision parameters:

$$P(\beta|\tau, \alpha, \alpha_0, \beta_0) \sim \Gamma\left(\alpha_0 + \alpha V, \beta_0 + \sum_{v=1}^V (\tau_v)\right),$$

where $V$ is the length of the vector $\tau$ (i.e. number of traits analysed).

The inclusion in the model of traits that have no effect on the extinction rates may result in over-parameterization that can yield spurious results and false positives. We addressed this issue in two ways. First, our prior and hyper-prior settings (equations (3.3)–(3.5)) result in the Bayesian shrinkage of the $y$ parameters around 0, therefore shrinking the multipliers $m$ toward equal values (i.e. no effects of a trait on extinction). Specifically, as $\beta$ shrinks to smaller values (as determined by

the exponential hyper-prior), the precision parameters $\tau$ tend to become larger (i.e. smaller variance), therefore constraining the $y$ parameters to values close to 0. Second, we implemented a Bayesian variable selection algorithm where each trait $v$ is assigned an indicator $I_v$ which can take a value of 0 or 1 (see Silvestro *et al.* [17] for a similar implementation). The indicators are used as multipliers to the $y$ vectors so that when $I_v = 0$ all resulting $m$ are equal (equation (3.2)). The indicators are treated as unknown parameters and estimated from the data through MCMC. We set the prior probability to $P(I = 1) = 0.05$ (therefore placing 0.95 probability on the absence of an effect). Under this prior a posterior frequency of $I = 1$ greater than 51.4% is considered as significant (2 log Bayes factor greater than 6 [69]).

We ran 10 replicated analyses using the resampled longevities of species inferred in the PyRate analyses of speciation, extinction, and preservation described above. For each analysis we ran 2 million MCMC iterations sampling every 1000 iterations. After combining all the output files, we computed the posterior frequency of each indicator to assess which traits were estimated to have a significant effect on extinction. We summarized any trait-dependent effects based on the sampled multipliers $m$. Because extinction is measured as a rate, i.e. a per-species value, the estimated relative effects on extinction is independent of the number of species in a state.

The results from the MTE model did not appear to be affected by species occupancy (a proxy of rarity; see electronic supplementary material, figure S8).

### (i) Functional redundancy in the multi-trait-dependent extinction model

We used the results of the functional diversity analysis to assess the role of functional redundancy in extinctions. To do so, we categorized species based on their level of redundancy by quantifying the number of species per FE. Accordingly, a species featuring a unique trait combination will have the lowest redundancy (i.e. will be uniquely occupying its FE), whereas a species that has the same combination of traits as many other species is considered as highly redundant (i.e. will occupy an FE along with other species). To test if and how the differences in functional redundancy are linked to extinction rates, we (i) translated functional redundancy from FEs to species, (ii) partitioned

the log10-transformed functional redundancy of each species into five discrete categories, and (iii) used the extinction models described below to assess whether redundancy significantly affected extinction rates. The log10-transformed species-level functional redundancy (logFR) was arbitrarily categorized based as follows:

1. unique (logFR = 0; i.e. only one species occupies its FE)
2. very low (logFR < 0.5; i.e. species share FE with up to 3 species)
3. low (0.5 < logFR < 1; i.e. sharing FE with up to 10 species)
4. high (1 < logFR < 1.5; i.e. sharing FE with up to 32 species)
5. very high (logFR > 1.5; i.e. sharing FE with >32 species).

Data accessibility. Additional methods, tables, figures, and references supporting this article have been uploaded as part of the online electronic supplementary material.

Authors' contributions. A.H. compiled the data; C.P., D.S., and A.Z. analysed the data; D.S. and X.M. developed the MTE model; C.P., C.D.B., and D.S. led the writing; all co-authors provided ideas and various contributions.

Competing interests. The authors declare no competing interests.

Funding. C.P was partially funded by the Alexander von Humboldt Foundation and the Federal Ministry for Education and Research (Germany). C.D.B was funded by the Swedish Research Council (2017-04980). D.S. received funding from the Swiss National Science Foundation (PCEFP3_187012; FN-1749) and from the Swedish Research Council (VR: 2019-04739). X.M. was supported by the Swiss National Science Foundation (P2GEP2-178032). A.A. was supported by the Swedish Foundation for Strategic Research, the Swedish Research Council, the Knut and Alice Wallenberg Foundation, and the Royal Botanic Gardens, Kew. This research has received funding from the European Union's Horizon 2020 research and innovation programme under the Marie Skłodowska-Curie grant agreement no. 663830.

Acknowledgements. We thank the curators, staff, and field collectors of the data presented in the electronic supplementary dataset, particularly of those institutions funded through the National Science Foundation Advancing Digitization of Biological Collections Paleo-Niches Thematic Collections Network accessed from iDigBio (NSF# 1206757, 1206769). We also thank J.N. Griffin and S. Faurby for providing valuable feedback. Finally, we are grateful for the insightful reviews from P. Novack-Gottshall and three anonymous referees that greatly improved the manuscript.

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
