## [Reviewer comments · Proceedings of the Royal Society B: Biological Sciences]

Review History

RSPB-2019-2099.R0 (Original submission)

Review form: Reviewer 1

Recommendation

Major revision is needed (please make suggestions in comments)

Scientific importance: Is the manuscript an original and important contribution to its field?

Good

General interest: Is the paper of sufficient general interest?

Good

Quality of the paper: Is the overall quality of the paper suitable?

Excellent

Is the length of the paper justified?

Yes

Should the paper be seen by a specialist statistical reviewer?

Yes

Do you have any concerns about statistical analyses in this paper? If so, please specify them explicitly in your report.

No

It is a condition of publication that authors make their supporting data, code and materials available - either as supplementary material or hosted in an external repository. Please rate, if applicable, the supporting data on the following criteria.

Is it accessible?

Yes

Is it clear?

Yes

Is it adequate?

Yes

Do you have any ethical concerns with this paper?

No

Comments to the Author

This is the report for «Redundancy buffers functional diversity against extinction in Caribbean mollusks» by Bacon et al. In this study, the authors analyze an impressive dataset of 44,000 mollusks fossil occurrences in the Caribbean region and couple it to a trait dataset using an advanced Bayesian approach. The authors aim to study diversity changes from the Neogene to the Quaternary and assess whether species extinctions was linked to the species' traits (identity and redundancy). The authors find that extinctions were not occurring randomly across species but that extinctions rates were higher for species that tended to be redundant (their trait characteristics are shared by other species). As a result, functional richness (calculated as the volume in the trait space, i.e. trait richness) stayed significantly high (higher than expected by chance) after extinction events. This suggest that ecosystem with high functional redundancy might be more resilient to extinction events.

The manuscript of Bacon et al. brings novel insights into the mechanism of species extinction and is extremely relevant under the ongoing climate crisis to assess which species are more likely at risk of extinction (i.e. redundant species). I enjoyed reading this high-quality manuscript and I believe it will be of wide interests to the readers of RSPB.

I have however some comments and suggestions for clarifications. I would especially like the authors to expand the discussion, which is for now minimal. Among other I would like them to discuss the potential limitation of the used traits (e.g. if they had selected other traits & categories or added more traits how could this have impacted the functional redundancy results?), and of the methods (fossilization rate, whether there is potential trait-based bias in encounter rate for e.g. smaller species). Discuss also why some trait categories are linked to a higher extinction rate as well as on the relevance of the results. I also have some comments on the method used for doing the null model (is the frequency of occurrence taken into account?) and the calculation of the species-level redundancy, but this might just be a misunderstanding due to some lack of precision in the M&M. Please find more details on these issued below.

Introduction:

L101-103: I don't think the questions formulated here really reflect the main analyses of the study, please reformulate. Question 1 & 2 appear to be the same (functional diversity included in "diversity") and 3 not targeted particular "species" but particular "traits".

M&M:

Is the fossilization rate the same between the different mollusks class? Can some of the studied traits have some influence on the fossilization or fossil encounter rate? And what could be the impact of this potential bias on the results? If I understood correctly, the authors did take into account the potential bias in their Bayesian approach, but it would be good to have this potential limitation stated much more explicitly to build more trust on the results.

L251-253: this is not really clear. So each record dated to a specific geological epoch was assigned as being present during all of this epoch? While in L270 the authors mention “the observed temporal range between first and last appearances”. Understanding how the different species were assigned as present/absent in the different time bin (1Myr) is crucial to understand well the method, so I will urge the authors to present more clearly the different sequential steps that were required to obtain the species*time occurrence matrix (L298).

L266. Please explain more what the preservation biases refers to and how it is taken into account.

L290: state the 5 traits. What is the criteria for differentiating deposit feeders and detrital? Deposit feeders (and suspension feeders) could feed on detritus?

L290: rephrase. “structure” is often used to refer to species identity, as you are using traits here it might be better to choose word such as “characterize the ecological niche” (and delete the last part of the sentence “functional diversity”).

L319: the implementation of the null model needs more details. The null model was done per time bin and not across time bin? If across time bin, did you also take into account the different species frequencies when doing the randomization? When doing community null model the randomization should not only keep species richness constant (keeping the same number of species for each time bin and randomly selecting species from the species pool) but also taking into account the frequency of occurrence of species (species that are occurring in many time bins are more likely to be randomly selected than species in few time bins).

L441: Is the species-level functional redundancy estimated using the entire species pool (irrespective of time)? A species might be redundant when aggregated for the complete studied period but be unique during a time-bin? To be in line with the question addressed here, it might be better to calculate species-level functional redundancy as the average functional redundancy of the species across time-bins, although I do not think this would change much the results.

Results & Discussion:

L278: I didn't understand all the resort of this analysis, but from the Fig S5 it seems high only from -5Ma?

I would like to see the discussion expanded especially on the traits identity that appears to have high effect on extinction rates, and on how these results can inform on the currently threaten ecosystems e.g. resilience and management (building up on what is stated in L99 “could potentially help addressing current and future extinction scenarios).

L204: This is somehow an unexpected result, as large species are often considered more vulnerable and thus prone to extinctions. If I understood correctly, the argument of the authors is that this might be because there is a higher number of small species (higher redundancy). I am wondering whether it would not also be interesting to standardize the extinctions per number of species in each trait category i.e. proportionally would the extinction rate per size class then changes and become higher for larger species? I think it is important to make a distinction between absolute/relative extinction and I would at least like to have more discussion on this size-based extinction rate as it is not the results one would especially expect.

L181. In Fig 3 (and Fig S6) “suspension” feeder species does not appear to have an especially high extinction rate, remove. In the introduction, L82 you state that other studied demonstrated a higher extinction rate for suspension feeding bivalves and predatory gastropods, which you do not find here. What could be a reason for the observed difference? Please discuss this result more. Also, what is the difference between figure S6 and 3?

L225: think of rephrasing, as you do not explicitly analyzed the temporal changes in diversity vs environmental change (e.g. “coincide with” instead of “was related”).

Others:

The key word mention “Isthmus of Panama”. It is not referred to anywhere else in the text.

Consider changing the location mention

In Fig S1. Deposit is missing from the diet histogram.

Fig S2. Line 3, what does CI 10-14 refers to?

Table S2. "chemo or photo symbiotic" diet are not on the histogram fig S1

Review form: Reviewer 2 (Philip Novack-Gottshall)

Recommendation

Accept with minor revision (please list in comments)

Scientific importance: Is the manuscript an original and important contribution to its field?

Good

General interest: Is the paper of sufficient general interest?

Excellent

Quality of the paper: Is the overall quality of the paper suitable?

Good

Is the length of the paper justified?

Yes

Should the paper be seen by a specialist statistical reviewer?

Yes

Do you have any concerns about statistical analyses in this paper? If so, please specify them explicitly in your report.

No

It is a condition of publication that authors make their supporting data, code and materials available - either as supplementary material or hosted in an external repository. Please rate, if applicable, the supporting data on the following criteria.

Is it accessible?

Yes

Is it clear?

Yes

Is it adequate?

Yes

Do you have any ethical concerns with this paper?

No

Comments to the Author

See attached (See Appendix A).

Review form: Reviewer 3

Recommendation

Major revision is needed (please make suggestions in comments)

Scientific importance: Is the manuscript an original and important contribution to its field?

Acceptable

General interest: Is the paper of sufficient general interest?

Good

Quality of the paper: Is the overall quality of the paper suitable?

Acceptable

Is the length of the paper justified?

Yes

Should the paper be seen by a specialist statistical reviewer?

Yes

Do you have any concerns about statistical analyses in this paper? If so, please specify them explicitly in your report.

Yes

It is a condition of publication that authors make their supporting data, code and materials available - either as supplementary material or hosted in an external repository. Please rate, if applicable, the supporting data on the following criteria.

Is it accessible?

Yes

Is it clear?

Yes

Is it adequate?

No

Do you have any ethical concerns with this paper?

No

Comments to the Author

This paper analyses a large dataset for extinction patterns in taxonomic and functional diversity in West Atlantic mollusks during the Neogene. Like many authors, they find an extinction pulse in the late Neogene, and as reported at several other, larger extinction events, see modest loss of functional diversity relative to taxonomic losses at this time.

The ms states that "over the last three million years, Caribbean mollusks went through a period of high extinction driven by oceanographic changes." This is not strictly equivalent to the statement in the text that the maximum extinction is "between 3.2 and 2.6 Ma (figure 2b)". More importantly it does not match the figure, which clearly shows the major pulse in extinction rates to be in the Early Pliocene, from 3.5 or 4.0 Myr ago to a little more than 5 Myr (in binned terms, the Zanclean, 3.6-5.3 Ma). This is not a trivial issue, as the explanation for the observed patterns hinges in part on the timing of geological events. They further note the suggestion "that the extinction of mollusks presented a million-year temporal lag between environmental change and detectable extinction, becoming extinct between 2 and 1 Ma," an even larger discrepancy with

their data; this contradiction is also never addressed.

It has long been known (for example Foote 2003, 2010, but many other sources) that artificial origination pulses occur in intervals following times of poorer sampling (just as artificial extinction pulses precede times of poor sampling). This sampling effect could explain the supposed high rate of origination in the Aquitanian relative to later intervals, given that the preceding Chattian interval has ~430 occurrences (vs 660 occurrences in the following Aquitanian interval and ~5000 occurrences in the Burdigalian interval after that); evidently PyRate is sensitive to some types variations in sampling/preservation but not others, and clearly this point needs to be clarified; perhaps it accounts for the different results obtained by Crouch & Clarke 2019 when analyzing data using PyRate and the TRiPS method of Starfelt & Liow 2016 (Phil Trans). In a related point, a clear explanation is also needed for the estimate of constant preservation rates through time from ~23 Ma to ~5 Ma, given that the number of occurrences per bin within that interval vary from 662 to 5500 or 6000, depending on how composite time bins are treated. How, in fact, were occurrences in composite time bins treated (Burdigalian-Langhian, etc.)? Was every occurrence counted once in each of the two time bins, were they apportioned in some way between the two time bins, were they counted in the continuous time series as intermediate in age between the two bins, or as present in every one-Myr increment through the composite bin? The number of occurrences in such bins ranges from ~160 to more than 6700, so this is not a trivial analytical decision, but it is not referred to in the text or supplement.

A number of results rely on conversion of the time-binned data to continuous timeseries. For example, the authors mention a low-origination interval extending 5-10 Ma, followed by an origination pulse at 4-3.4 Ma. How was this pulse constrained so tightly when the Early Pliocene data occur in a bin dated at 3.6-5.3 Ma? Similarly, the ms needs to clarify how the speciation pulse shown as a plateau from 20-22 Ma was generated when the raw data fall within the Aquitanian stage, which starts at 23.7 Ma, when there is a trough shown in the speciation rates. These issues probably point to the need to explain how they converted their time-binned data to continuous timeseries as shown in the figures, given the potential for biases introduced by time-binned data, particularly with unequal time bins as used here (ranging from 0.9 Ma to 6.6 Ma in length) (see Foote 2000, Alroy 2010, and many other references). Were the biases tested for, or otherwise shown to be minor effects? The paper mentions "time windows", are these equal to the stage-level bins tabulated in the Supplement, or are these 1-Myr increments, as implied in Supplement Tables S1 and S3.

In the .csv dataset, Column G gives the geo.region for all localities, including those whose latitudes correspond to South Carolina and Georgia, as "Caribbean". Authors such as Woodring, Petuch, and Vermeij have shown that North Carolina-Florida was a distinct biogeographic province through most or all of the study interval, with a different fauna and spatio-temporal dynamics from the core Caribbean fauna (sometimes termed the Caloosahatchian and Gatunian provinces respectively). Some demonstration is needed that merging these two provinces in the present study is analytically appropriate.

The ms concludes that "molluscan communities experienced major diversity changes during the last 23 million years" but given that the analyses lump the entire region from Venezuela and Panama to Texas and South Carolina, no direct conclusions can be made at the community level. Although extinction did likely reduce alpha-diversity, no account is taken of changes in beta diversity, which might also contribute to net regional diversity increases and decreases, or as noted above, there no consideration of different diversity changes in the two provinces lumped together here.

It is difficult to evaluate spatial pattern in the data. Figure 1 is not very informative in that it does not indicate how the occurrences are distributed through time. It is striking how many of these points are in cool colors, indicating that they represent only 1-~10 occurrences. Does spatial coverage, defined by locations yielding some minimum number of occurrences, change significantly through the timeseries? If so, how was that taken into account? The ms attributable

increasing taxonomic and functional diversity to the increase in available habitat types, but all of their habitat categories are present in all of their time bins, and there is no treatment of the alternative sampling hypothesis, i.e. amount of each habitat type sampled in a time bin. Quantifying spatial coverage per time bin is important here.

Regarding the conclusions, Edie et al. 2018 PNAS, surprisingly not cited here, also found a mismatch between taxonomic and functional extinctions, at two major extinction events, the end-Permian and end-Cretaceous; given these and other prior results (Erwin et al. 1987 is the pioneering study that also found minor functional losses despite major taxonomic extinction, and also is not cited), is it surprising that a study of a smaller, regional extinction produced similar findings? Further, what is the actual mechanism for “selectivity against functionally redundant species”? I see that Edie et al. also discussed extinctions in these terms, and concentrated in the most richly occupied functional groups -- that paper certainly should be cited and the results here related to that study. Edie et al. outlined four hypotheses for the uneven extinction they also observed among functional groups, are any of those definitively rejected or supported by the analyses in the current ms?

Minor points

On Fig. 1, the data points should be larger. And it is not clear what the final phrase “and recent occurrence records are extant collections” is referring to.

It is difficult to navigate this .csv file, and an additional field should be added for placenames – minimally countries, and states for the larger countries such as Venezuela and the U.S.

The min-age for the Chattian is given as 23 Ma, but the max-age for the immediately overlying Aquitanian is given as 23.7 Ma, yielding an overlap of the two timebins. Did this distort numbers in the 23 Ma time bin? The time series in Figure 2 shows only the Miocene-Recent, where do the 431 Chattian records enter into the analyses?

What does “NA” in column A signify for the analyses? Were these 13,851 occurrences of 235 species omitted, or otherwise used differently from the occurrences that have numbers in column A? Were they sourced differently?

Review form: Reviewer 4

Recommendation

Major revision is needed (please make suggestions in comments)

Scientific importance: Is the manuscript an original and important contribution to its field?

Good

General interest: Is the paper of sufficient general interest?

Good

Quality of the paper: Is the overall quality of the paper suitable?

Good

Is the length of the paper justified?

Yes

Should the paper be seen by a specialist statistical reviewer?

No

Do you have any concerns about statistical analyses in this paper? If so, please specify them explicitly in your report.

No

It is a condition of publication that authors make their supporting data, code and materials available - either as supplementary material or hosted in an external repository. Please rate, if applicable, the supporting data on the following criteria.

Is it accessible?

Yes

Is it clear?

Yes

Is it adequate?

Yes

Do you have any ethical concerns with this paper?

No

Comments to the Author

The role of functional diversity has taken its rightful place in paleobiological studies, and this work by Bacon et al. takes it one step further by investigating the role of functional redundancy in extinction selectivity. This question has obvious implications not only for paleobiology, but also for conservation ecology and the ongoing mass extinction. The authors have compiled and augmented an impressive dataset on Caribbean mollusks spanning the last 23 million years, and they use a robust, probabilistic framework for estimating macroevolutionary rates and drawing inferences. Their results and conclusions come across as eminently sensible: functional diversity increases through time, and extinction rates are consistently higher in functionally redundant species, implying that selective extinction against functional redundancy enhances ecosystem resilience. Their results also suggest that competitive interactions mediate the effects of environmental drivers of extinction.

There is, however, one substantive issue that I believe the authors should address, and that is the role of abundance/rarity as a possible confounding factor in the association between functional traits and extinction. The trait distributions are presented in terms of richness per trait category, and one is left to wonder what those distributions look like in terms of occurrences (presumably, not vastly different, as there is no mention of standardized richness). In the case of the 'diet' trait, for example, the most frequent trait categories (e.g. predator, suspension) seem to have no relative effect on extinction (caveat: the deposit feeding category is missing from the histogram in Fig. S1). Is the relationship between a functional trait category and extinction decoupled from the abundance/rarity of species?

Addressing this question is important for two reasons:

(1) Confounding. Rarity may impart greater extinction vulnerability. If we allocate species to trait categories, and most species are rare (in a relative sense), then some categories may contain mostly or only rare species. If that were the case, then a strong association between the trait category and extinction may be due to rarity, rather than the trait itself. Conceivably, this confounding could propagate into the functional redundancy-extinction association as well: if most species are rare, then we might see a kind of "functional central limit theorem" by which there are more ways to be rare than to be common and thus more opportunities for shared functional traits.

(2) Ecological implications. If rare species tend to be redundant and have little effect on ecosystem functioning, then their extinction may not be a cause for concern, it may even confer resilience. On the other hand, if the redundancy-extinction link is found to be decoupled from rarity, or if rare species tend to be functionally unique, they may instead buffer ecosystems (e.g. <https://doi.org/10.1371/journal.pbio.1001569>). Either way, such a finding in deep-time data

would have rich implications for conservation.

Given fossil occurrence data, relative species abundance can be assessed using simple measures of regional occupancy (e.g. by gridding the occurrence data). With an occupancy-based measure of abundance (i.e. common and widespread, not necessarily locally abundant), it would be feasible to test the possible role of rarity in the relationship between functional traits and extinction. This test should not involve an unreasonable amount of additional work for this manuscript. The potential role of rarity in the redundancy-extinction relationship could possibly also be investigated by numerical experiment: given a log-normal species abundance distribution, assign species to arbitrary trait categories, and see if redundancy is associated with rarity, regardless of trait.

Unless my suggestions can be dismissed on conceptual/logical grounds, I believe adding a rarity dimension would enrich the ecological interpretation and widen the implications of this study.

Minor comments:

Figure S1: y-axis labels are missing, and one of the trait categories is missing from three of the panels. The caption repeats the information in Table S2, and I suggest showing trait categories directly on the x-axis as tick labels to enhance readability.

Check the order of appearance of Supplementary figure references in the main text.

Figure S6 seems identical to the final panel in Figure 3b. Redundancy -> extinction.

No need to repeat the numbers in line 262 and line 240.

Lines 278-9: Please explain how the pattern in Fig S5 can be described as "overall high".

Decision letter (RSPB-2019-2099.R0)

13-Dec-2019

Dear Ms Pimiento:

I am writing to inform you that your manuscript RSPB-2019-2099 entitled "Redundancy buffers functional diversity against extinction in Caribbean mollusks" has, in its current form, been rejected for publication in Proceedings B.

This action has been taken on the advice of referees, who have recommended that substantial revisions are necessary. With this in mind we would be happy to consider a resubmission, provided the comments of the referees are fully addressed. However please note that this is not a provisional acceptance.

Please find below the comments made by the referees, not including confidential reports to the

Editor, which I hope you will find useful. If you do choose to resubmit your manuscript, please upload the following:

Please note that this decision may (or may not) have taken into account confidential comments.

In your revision process, please take a second look at how open your science is; our policy is that all data involved with the study should be made openly accessible-- see: <https://royalsociety.org/journals/ethics-policies/data-sharing-mining/>
Insufficient sharing of data can delay or even cause rejection of a paper.

Sincerely,
Professor John Hutchinson, Editor
mailto: proceedingsb@royalsociety.org

Associate Editor

Board Member: 1

Comments to Author:

The authors use an impressive dataset and a sophisticated suite of analyses to address how functional diversity has changed over the last 23 million years using Western Atlantic molluscs. The reviewers and I generally find the study of value and interest to the broad readership of PRSB. The study aims of identifying extinction mechanisms are particularly timely given the current high rate of biodiversity loss.

However, a number of concerns were raised regarding the manuscript and accompanying analyses, which the authors should find useful when revising. All reviewers, for example, raised concerns about the clarity of the methods as presented, such that a revised version of the manuscript should ensure each analysis is reproducible. All reviewers also questioned how spatial coverage and sampling could affect patterns obtained, and I encourage the authors to consider sensitivity analyses to address these concerns.

Reviewer 1 mentioned that the Discussion needs expansion, and that the study aims/questions (Intro) need to be reformulated to reflect the nature of the analyses conducted. Reviewer 2 raised a concern regarding the lack of statistical tests to analyse the major patterns reported in Figure 2, which I echo, and questioned some of the patterns the authors suggest based on the data presented. The authors should consider whether the diversification trends truly reflect competition or could simply result from a random exploration of ecospace. Reviewer 3 mentioned that the pulse of extinction cited at 3.2 to 2.6 Ma does not correspond with the data presented, which instead indicates a major pulse of extinction at 3.5 to 4 Ma. Both Reviewer 3 and I also question the inclusion of areas such as North Carolina to Florida in a definition of 'Caribbean', and some justification of this choice is warranted. Finally, Reviewer 4 wondered about the role of abundance as a possible confounding factor in the association between functional traits and extinction.

These are not insignificant issues, and they will need to be addressed in order to be considered suitable for publication.

Reviewer(s)' Comments to Author:

Referee: 1

Comments to the Author(s)

This is the report for «Redundancy buffers functional diversity against extinction in Caribbean mollusks» by Bacon et al. In this study, the authors analyze an impressive dataset of 44,000 mollusk fossil occurrences in the Caribbean region and couple it to a trait dataset using an advanced Bayesian approach. The authors aim to study diversity changes from the Neogene to the Quaternary and assess whether species extinctions were linked to the species' traits (identity and redundancy). The authors find that extinctions were not occurring randomly across species but that extinction rates were higher for species that tended to be redundant (their trait characteristics are shared by other species). As a result, functional richness (calculated as the volume in the trait space, i.e. trait richness) stayed significantly high (higher than expected by chance) after extinction events. This suggests that an ecosystem with high functional redundancy might be more resilient to extinction events.

The manuscript of Bacon et al. brings novel insights into the mechanism of species extinction and is extremely relevant under the ongoing climate crisis to assess which species are more likely at risk of extinction (i.e. redundant species). I enjoyed reading this high-quality manuscript and I believe it will be of wide interest to the readers of RSPB.

I have however some comments and suggestions for clarifications. I would especially like the authors to expand the discussion, which is for now minimal. Among other I would like them to discuss the potential limitation of the used traits (e.g. if they had selected other traits & categories or added more traits how could this have impacted the functional redundancy results?), and of the methods (fossilization rate, whether there is potential trait-based bias in encounter rate for e.g. smaller species). Discuss also why some trait categories are linked to a higher extinction rate as well as on the relevance of the results. I also have some comments on the method used for doing the null model (is the frequency of occurrence taken into account?) and the calculation of the species-level redundancy, but this might just be a misunderstanding due to some lack of precision in the M&M. Please find more details on these issues below.

Introduction:

L101-103: I don't think the questions formulated here really reflect the main analyses of the study, please reformulate. Question 1 & 2 appear to be the same (functional diversity included in "diversity") and 3 not targeted particular "species" but particular "traits".

M&M:

Is the fossilization rate the same between the different mollusk classes? Can some of the studied traits have some influence on the fossilization or fossil encounter rate? And what could be the impact of this potential bias on the results? If I understood correctly, the authors did take into account the potential bias in their Bayesian approach, but it would be good to have this potential limitation stated much more explicitly to build more trust on the results.

L251-253: this is not really clear. So each record dated to a specific geological epoch was assigned as being present during all of this epoch? While in L270 the authors mention "the observed temporal range between first and last appearances". Understanding how the different species were assigned as present/absent in the different time bin (1Myr) is crucial to understand well the method, so I will urge the authors to present more clearly the different sequential steps that were required to obtain the species*time occurrence matrix (L298).

L266. Please explain more what the preservation biases refers to and how it is taken into account.

L290: state the 5 traits. What is the criteria for differentiating deposit feeders and detrital? Deposit feeders (and suspension feeders) could feed on detritus?

L290: rephrase. "structure" is often used to refer to species identity, as you are using traits here it might be better to choose word such as "characterize the ecological niche" (and delete the last part of the sentence "functional diversity").

L319: the implementation of the null model needs more details. The null model was done per time bin and not across time bin? If across time bin, did you also take into account the different species frequencies when doing the randomization? When doing community null model the randomization should not only keep species richness constant (keeping the same number of species for each time bin and randomly selecting species from the species pool) but also taking into account the frequency of occurrence of species (species that are occurring in many time bins are more likely to be randomly selected than species in few time bins).

L441: Is the species-level functional redundancy estimated using the entire species pool (irrespective of time)? A species might be redundant when aggregated for the complete studied period but be unique during a time-bin? To be in line with the question addressed here, it might be better to calculate species-level functional redundancy as the average functional redundancy of the species across time-bins, although I do not think this would change much the results.

Results & Discussion:

L278: I didn't understand all the resort of this analysis, but from the Fig S5 it seems high only from -5Ma?

I would like to see the discussion expanded especially on the traits identity that appears to have high effect on extinction rates, and on how these results can inform on the currently threaten ecosystems e.g. resilience and management (building up on what is stated in L99 "could potentially help addressing current and future extinction scenarios).

L204: This is somehow an unexpected result, as large species are often considered more vulnerable and thus prone to extinctions. If I understood correctly, the argument of the authors is that this might be because there is a higher number of small species (higher redundancy). I am wondering whether it would not also be interesting to standardize the extinctions per number of species in each trait category i.e. proportionally would the extinction rate per size class then changes and become higher for larger species? I think it is important to make a distinction between absolute/relative extinction and I would at least like to have more discussion on this size-based extinction rate as it is not the results one would especially expect.

L181. In Fig 3 (and Fig S6) "suspension" feeder species does not appear to have an especially high extinction rate, remove. In the introduction, L82 you state that other studied demonstrated a higher extinction rate for suspension feeding bivalves and predatory gastropods, which you do not find here. What could be a reason for the observed difference? Please discuss this result more. Also, what is the difference between figure S6 and 3?

L225: think of rephrasing, as you do not explicitly analyzed the temporal changes in diversity vs environmental change (e.g. "coincide with" instead of "was related").

Others:

The key word mention "Isthmus of Panama". It is not referred to anywhere else in the text.

Consider changing the location mention

In Fig S1. Deposit is missing from the diet histogram.

Fig S2. Line 3, what does CI 10-14 refers to?

Table S2. "chemo or photo symbiotic" diet are not on the histogram fig S1

Referee: 2

Comments to the Author(s)

See attached.

Referee: 3

Comments to the Author(s)

This paper analyses a large dataset for extinction patterns in taxonomic and functional diversity in West Atlantic mollusks during the Neogene. Like many authors, they find an extinction pulse in the late Neogene, and as reported at several other, larger extinction events, see modest loss of functional diversity relative to taxonomic losses at this time.

The ms states that “over the last three million years, Caribbean mollusks went through a period of high extinction driven by oceanographic changes.” This is not strictly equivalent to the statement in the text that the maximum extinction is “between 3.2 and 2.6 Ma (figure 2b)”. More importantly it does not match the figure, which clearly shows the major pulse in extinction rates to be in the Early Pliocene, from 3.5 or 4.0 Myr ago to a little more than 5 Myr (in binned terms, the Zanclean, 3.6-5.3 Ma). This is not a trivial issue, as the explanation for the observed patterns hinges in part on the timing of geological events. They further note the suggestion “that the extinction of mollusks presented a million-year temporal lag between environmental change and detectable extinction, becoming extinct between 2 and 1 Ma,” an even larger discrepancy with their data; this contradiction is also never addressed.

It has long been known (for example Foote 2003, 2010, but many other sources) that artificial origination pulses occur in intervals following times of poorer sampling (just as artificial extinction pulses precede times of poor sampling). This sampling effect could explain the supposed high rate of origination in the Aquitanian relative to later intervals, given that the preceding Chattian interval has ~430 occurrences (vs 660 occurrences in the following Aquitanian interval and ~5000 occurrences in the Burdigalian interval after that); evidently PyRate is sensitive to some types variations in sampling/preservation but not others, and clearly this point needs to be clarified; perhaps it accounts for the different results obtained by Crouch & Clarke 2019 when analyzing data using PyRate and the TRiPS method of Starfelt & Liow 2016 (Phil Trans). In a related point, a clear explanation is also needed for the estimate of constant preservation rates through time from ~23 Ma to ~5 Ma, given that the number of occurrences per bin within that interval vary from 662 to 5500 or 6000, depending on how composite time bins are treated. How, in fact, were occurrences in composite time bins treated (Burdigalian-Langhian, etc.)? Was every occurrence counted once in each of the two time bins, were they apportioned in some way between the two time bins, were they counted in the continuous time series as intermediate in age between the two bins, or as present in every one-Myr increment through the composite bin? The number of occurrences in such bins ranges from ~160 to more than 6700, so this is not a trivial analytical decision, but it is not referred to in the text or supplement.

A number of results rely on conversion of the time-binned data to continuous timeseries. For example, the authors mention a low-origination interval extending 5-10 Ma, followed by an origination pulse at 4-3.4 Ma. How was this pulse constrained so tightly when the Early Pliocene data occur in a bin dated at 3.6-5.3 Ma? Similarly, the ms needs to clarify how the speciation pulse shown as a plateau from 20-22 Ma was generated when the raw data fall within the Aquitanian stage, which starts at 23.7 Ma, when there is a trough shown in the speciation rates. These issues probably point to the need to explain how they converted their time-binned data to continuous timeseries as shown in the figures, given the potential for biases introduced by time-binned data, particularly with unequal time bins as used here (ranging from 0.9 Ma to 6.6 Ma in length) (see Foote 2000, Alroy 2010, and many other references). Were the biases tested for, or otherwise shown to be minor effects? The paper mentions “time windows”, are these equal to the stage-level bins tabulated in the Supplement, or are these 1-Myr increments, as implied in Supplement Tables S1 and S3.

In the .csv dataset, Column G gives the geo.region for all localities, including those whose latitudes correspond to South Carolina and Georgia, as “Caribbean”. Authors such as Woodring, Petuch, and Vermeij have shown that North Carolina-Florida was a distinct biogeographic province through most or all of the study interval, with a different fauna and spatio-temporal

dynamics from the core Caribbean fauna (sometimes termed the Caloosahatchian and Gatunian provinces respectively). Some demonstration is needed that merging these two provinces in the present study is analytically appropriate.

The ms concludes that “molluscan communities experienced major diversity changes during the last 23 million years” but given that the analyses lump the entire region from Venezuela and Panama to Texas and South Carolina, no direct conclusions can be made at the community level. Although extinction did likely reduce alpha-diversity, no account is taken of changes in beta diversity, which might also contribute to net regional diversity increases and decreases, or as noted above, there no consideration of different diversity changes in the two provinces lumped together here.

It is difficult to evaluate spatial pattern in the data. Figure 1 is not very informative in that it does not indicate how the occurrences are distributed through time. It is striking how many of these points are in cool colors, indicating that they represent only 1~10 occurrences. Does spatial coverage, defined by locations yielding some minimum number of occurrences, change significantly through the timeseries? If so, how was that taken into account? The ms attributable increasing taxonomic and functional diversity to the increase in available habitat types, but all of their habitat categories are present in all of their time bins, and there is no treatment of the alternative sampling hypothesis, i.e. amount of each habitat type sampled in a time bin. Quantifying spatial coverage per time bin is important here.

Regarding the conclusions, Edie et al. 2018 PNAS, surprisingly not cited here, also found a mismatch between taxonomic and functional extinctions, at two major extinction events, the end-Permian and end-Cretaceous; given these and other prior results (Erwin et al. 1987 is the pioneering study that also found minor functional losses despite major taxonomic extinction, and also is not cited), is it surprising that a study of a smaller, regional extinction produced similar findings? Further, what is the actual mechanism for “selectivity against functionally redundant species”? I see that Edie et al. also discussed extinctions in these terms, and concentrated in the most richly occupied functional groups -- that paper certainly should be cited and the results here related to that study. Edie et al. outlined four hypotheses for the uneven extinction they also observed among functional groups, are any of those definitively rejected or supported by the analyses in the current ms?

Minor points

On Fig. 1, the data points should be larger. And it is not clear what the final phrase “and recent occurrence records are extant collections” is referring to.

It is difficult to navigate this .csv file, and an additional field should be added for placenames – minimally countries, and states for the larger countries such as Venezuela and the U.S.

The min-age for the Chattian is given as 23 Ma, but the max-age for the immediately overlying Aquitanian is given as 23.7 Ma, yielding an overlap of the two timebins. Did this distort numbers in the 23 Ma time bin? The time series in Figure 2 shows only the Miocene-Recent, where do the 431 Chattian records enter into the analyses?

What does “NA” in column A signify for the analyses? Were these 13,851 occurrences of 235 species omitted, or otherwise used differently from the occurrences that have numbers in column A? Were they sourced differently?

Referee: 4

Comments to the Author(s)

The role of functional diversity has taken its rightful place in paleobiological studies, and this work by Bacon et al. takes it one step further by investigating the role of functional redundancy in extinction selectivity. This question has obvious implications not only for paleobiology, but also for conservation ecology and the ongoing mass extinction. The authors have compiled and augmented an impressive dataset on Caribbean mollusks spanning the last 23 million years, and they use a robust, probabilistic framework for estimating macroevolutionary rates and drawing

inferences. Their results and conclusions come across as eminently sensible: functional diversity increases through time, and extinction rates are consistently higher in functionally redundant species, implying that selective extinction against functional redundancy enhances ecosystem resilience. Their results also suggest that competitive interactions mediate the effects of environmental drivers of extinction.

There is, however, one substantive issue that I believe the authors should address, and that is the role of abundance/rarity as a possible confounding factor in the association between functional traits and extinction. The trait distributions are presented in terms of richness per trait category, and one is left to wonder what those distributions look like in terms of occurrences (presumably, not vastly different, as there is no mention of standardized richness). In the case of the 'diet' trait, for example, the most frequent trait categories (e.g. predator, suspension) seem to have no relative effect on extinction (caveat: the deposit feeding category is missing from the histogram in Fig. S1). Is the relationship between a functional trait category and extinction decoupled from the abundance/rarity of species?

Addressing this question is important for two reasons:

(1) Confounding. Rarity may impart greater extinction vulnerability. If we allocate species to trait categories, and most species are rare (in a relative sense), then some categories may contain mostly or only rare species. If that were the case, then a strong association between the trait category and extinction may be due to rarity, rather than the trait itself. Conceivably, this confounding could propagate into the functional redundancy-extinction association as well: if most species are rare, then we might see a kind of "functional central limit theorem" by which there are more ways to be rare than to be common and thus more opportunities for shared functional traits.

(2) Ecological implications. If rare species tend to be redundant and have little effect on ecosystem functioning, then their extinction may not be a cause for concern, it may even confer resilience. On the other hand, if the redundancy-extinction link is found to be decoupled from rarity, or if rare species tend to be functionally unique, they may instead buffer ecosystems (e.g. <https://doi.org/10.1371/journal.pbio.1001569>). Either way, such a finding in deep-time data would have rich implications for conservation.

Given fossil occurrence data, relative species abundance can be assessed using simple measures of regional occupancy (e.g. by gridding the occurrence data). With an occupancy-based measure of abundance (i.e. common and widespread, not necessarily locally abundant), it would be feasible to test the possible role of rarity in the relationship between functional traits and extinction. This test should not involve an unreasonable amount of additional work for this manuscript. The potential role of rarity in the redundancy-extinction relationship could possibly also be investigated by numerical experiment: given a log-normal species abundance distribution, assign species to arbitrary trait categories, and see if redundancy is associated with rarity, regardless of trait.

Unless my suggestions can be dismissed on conceptual/logical grounds, I believe adding a rarity dimension would enrich the ecological interpretation and widen the implications of this study.

Minor comments:

Figure S1: y-axis labels are missing, and one of the trait categories is missing from three of the panels. The caption repeats the information in Table S2, and I suggest showing trait categories directly on the x-axis as tick labels to enhance readability.

Check the order of appearance of Supplementary figure references in the main text.

Figure S6 seems identical to the final panel in Figure 3b. Redundancy -> extinction.

No need to repeat the numbers in line 262 and line 240.

Lines 278-9: Please explain how the pattern in Fig S5 can be described as "overall high".

Author's Response to Decision Letter for (RSPB-2019-2099.R0)

See Appendix B.

RSPB-2020-1162.R0

Review form: Reviewer 4

Recommendation

Major revision is needed (please make suggestions in comments)

Scientific importance: Is the manuscript an original and important contribution to its field?

Excellent

General interest: Is the paper of sufficient general interest?

Excellent

Quality of the paper: Is the overall quality of the paper suitable?

Good

Is the length of the paper justified?

Yes

Should the paper be seen by a specialist statistical reviewer?

No

Do you have any concerns about statistical analyses in this paper? If so, please specify them explicitly in your report.

Yes

It is a condition of publication that authors make their supporting data, code and materials available - either as supplementary material or hosted in an external repository. Please rate, if applicable, the supporting data on the following criteria.

Is it accessible?

Yes

Is it clear?

Yes

Is it adequate?

Yes

Do you have any ethical concerns with this paper?

No

Comments to the Author

Again, I wish to commend the authors on an overall solid and impressive study. I am very pleased to see that the authors have followed up on my suggestion to compute simple species occupancy on a spatial grid, to assess the potential influence of abundance/rarity on their results. Unfortunately, however, the authors' assessment and dismissal of any effect of rarity is not very convincing and makes me more curious about the potential role of rarity.

I hasten to add that a link between rarity and redundancy/extinction could be important for understanding ecological mechanisms, it is not simply a matter of statistical artifact or bias. Indeed, the authors concede in their response letter that this is an important issue (although not mentioned anywhere in the main text of their revised manuscript).

The authors address this issue by generating box plots of occupancy for each factor/trait found to have a significant effect on extinction (Figure S8). I suggested simple occupancy as a proxy for abundance due to the near universality of a positive abundance-occupancy relationship. The authors also include analogous box plots using occurrences, but unlike occupancy, raw occurrences are not a valid proxy for abundance. In other words, only the right-hand panels of Figure S8 are relevant for testing the effect of rarity.

In their description of the box-plot graphics (Fig. S8), the authors emphasize that the factor/trait categories found to be selected against (highlighted in red) include not only rare species, but also abundant species. From this observation they conclude that rarity did not affect their extinction selectivity results. In their words (Supplement p. 6): "We found that species abundances did not confound the effects of functional traits and redundancy on extinction because extinction was affected by trait categories that had both rare, or abundant species. Crucially, this is true for the case of redundancy, which is the main factor affected by extinction in our study."

However, what is most striking in the box plots (Fig. S8, right panels only) is that four out of the six extinction-related categories have the lowest observed occupancy values. Out of the 17 categories considered, only four have 75% of species with occupancy of 3 grid cells or less and a median occupancy of a single grid cell. Those four are all among the six that have a significant effect on extinction, including the most prominent category of 'very high redundancy'. Conversely, the categories with the highest occupancy values (as indicated by the 75th percentile top of the IQR box) are all among the categories with the weakest effect on extinction (cf. Fig. 2; including 'very low redundancy', 'shoreface', and 'very large/large'). It is not at all clear how these box plots support the authors' conclusion that abundance/rarity did not affect the extinction selectivity. Given the size of the data set, small but consistent differences in abundance distribution may very well play a role in the extinction selectivity, and the authors should not leave room for doubt.

With the overwhelming majority of species being rare (low occupancy), visual inspection of highly skewed box plots may not be the most useful tool for assessing if/how rarity affects the results of their analysis. In light of the authors' otherwise sophisticated approach to probabilistic inference, I believe they are in a good position to perform a more rigorous sensitivity analysis on the effect of abundance/rarity. I don't have personal experience with the specific methods used by the authors, so I don't know what would be the best approach, but here are some tentative suggestions:

i) Is it possible to perform a sensitivity analysis on the inclusion/exclusion of rare/abundant species, along the lines of the analysis the authors have performed on the inclusion/exclusion of traits (Fig. S7)? This could be problematic due to the numerical dominance of very rare species.

ii) Would it be possible to standardize the abundance (occupancy) distributions across trait categories by subsampling to ensure each category has the same representation of rare/abundant species? This would require Monte Carlo iteration, but should in principle give an idea of the robustness of the extinction selectivity results when variation in occupancy is neutralized.

iii) Why not include occupancy as a factor in the multi-trait dependent extinction model? Sure, it's a rough proxy, but so are the functional traits. Intuitively, this would be the ecologically most meaningful approach.

It is not my intention to make the authors jump through hoops. I would like to see this work published, and therefore I want to help make sure the conclusions are well supported. A more rigorous assessment/sensitivity test of the effect of abundance/rarity seems well within reach and would significantly strengthen the authors conclusions. As I argued in my first review, even if rarity is found to affect the redundancy-extinction link, the relationship between rarity and redundancy should advance our mechanistic understanding of the ecology of extinction and widen the implications of this study.

Decision letter (RSPB-2020-1162.R0)

11-Jun-2020

Dear Dr Bacon:

Your manuscript has now been peer reviewed and the reviews have been assessed by an Associate Editor. The reviewers' comments (not including confidential comments to the Editor) and the comments from the Associate Editor are included at the end of this email for your reference. As you will see, the reviewers and the Editors have raised some concerns with your manuscript and we would like to invite you to revise your manuscript to address them.

Only one reviewer still has comments and these focus on a need for there to be an analysis of rarity that more directly addresses their main concern. They will need to re-review this.

Research ethics:

Use of animals and field studies:

Please submit a copy of your revised paper within three weeks. If we do not hear from you within this time your manuscript will be rejected. If you are unable to meet this deadline please let us know as soon as possible, as we may be able to grant a short extension.

Best wishes,
Dr John Hutchinson, Editor
mailto: proceedingsb@royalsociety.org

Associate Editor Board Member

Comments to Author:

Bacon and colleagues have done an excellent job responding to the reviewer concerns and queries. I commend the authors for the thoroughness of their revision. I have very few concerns remaining, aside from the issue of rarity mentioned by the Reviewer. The Reviewer suggests a number of analyses that the authors should consider to broaden their results and explore further the relationship between rarity and redundancy-extinction link.

Reviewer(s)' Comments to Author:

Referee: 4

Comments to the Author(s).

Again, I wish to commend the authors on an overall solid and impressive study. I am very pleased to see that the authors have followed up on my suggestion to compute simple species occupancy on a spatial grid, to assess the potential influence of abundance/rarity on their results. Unfortunately, however, the authors' assessment and dismissal of any effect of rarity is not very convincing and makes me more curious about the potential role of rarity.

I hasten to add that a link between rarity and redundancy/extinction could be important for understanding ecological mechanisms, it is not simply a matter of statistical artifact or bias. Indeed, the authors concede in their response letter that this is an important issue (although not mentioned anywhere in the main text of their revised manuscript).

The authors address this issue by generating box plots of occupancy for each factor/trait found to have a significant effect on extinction (Figure S8). I suggested simple occupancy as a proxy for abundance due to the near universality of a positive abundance-occupancy relationship. The authors also include analogous box plots using occurrences, but unlike occupancy, raw occurrences are not a valid proxy for abundance. In other words, only the right-hand panels of Figure S8 are relevant for testing the effect of rarity.

In their description of the box-plot graphics (Fig. S8), the authors emphasize that the factor/trait categories found to be selected against (highlighted in red) include not only rare species, but also abundant species. From this observation they conclude that rarity did not affect their extinction selectivity results. In their words (Supplement p. 6): "We found that species abundances did not confound the effects of functional traits and redundancy on extinction because extinction was affected by trait categories that had both rare, or abundant species. Crucially, this is true for the case of redundancy, which is the main factor affected by extinction in our study."

However, what is most striking in the box plots (Fig. S8, right panels only) is that four out of the six extinction-related categories have the lowest observed occupancy values. Out of the 17 categories considered, only four have 75% of species with occupancy of 3 grid cells or less and a median occupancy of a single grid cell. Those four are all among the six that have a significant effect on extinction, including the most prominent category of 'very high redundancy'. Conversely, the categories with the highest occupancy values (as indicated by the 75th percentile top of the IQR box) are all among the categories with the weakest effect on extinction (cf. Fig. 2; including 'very low redundancy', 'shoreface', and 'very large/large'). It is not at all clear how these box plots support the authors' conclusion that abundance/rarity did not affect the extinction selectivity. Given the size of the data set, small but consistent differences in abundance

distribution may very well play a role in the extinction selectivity, and the authors should not leave room for doubt.

With the overwhelming majority of species being rare (low occupancy), visual inspection of highly skewed box plots may not be the most useful tool for assessing if/how rarity affects the results of their analysis. In light of the authors' otherwise sophisticated approach to probabilistic inference, I believe they are in a good position to perform a more rigorous sensitivity analysis on the effect of abundance/rarity. I don't have personal experience with the specific methods used by the authors, so I don't know what would be the best approach, but here are some tentative suggestions:

i) Is it possible to perform a sensitivity analysis on the inclusion/exclusion of rare/abundant species, along the lines of the analysis the authors have performed on the inclusion/exclusion of traits (Fig. S7)? This could be problematic due to the numerical dominance of very rare species.

ii) Would it be possible to standardize the abundance (occupancy) distributions across trait categories by subsampling to ensure each category has the same representation of rare/abundant species? This would require Monte Carlo iteration, but should in principle give an idea of the robustness of the extinction selectivity results when variation in occupancy is neutralized.

iii) Why not include occupancy as a factor in the multi-trait dependent extinction model? Sure, it's a rough proxy, but so are the functional traits. Intuitively, this would be the ecologically most meaningful approach.

It is not my intention to make the authors jump through hoops. I would like to see this work published, and therefore I want to help make sure the conclusions are well supported. A more rigorous assessment/sensitivity test of the effect of abundance/rarity seems well within reach and would significantly strengthen the authors conclusions. As I argued in my first review, even if rarity is found to affect the redundancy-extinction link, the relationship between rarity and redundancy should advance our mechanistic understanding of the ecology of extinction and widen the implications of this study.

Author's Response to Decision Letter for (RSPB-2020-1162.R0)

See Appendix C.

Decision letter (RSPB-2020-1162.R1)

29-Jun-2020

Dear Dr Bacon

I am pleased to inform you that your manuscript entitled "Selective extinction against redundant species buffers functional diversity" has been accepted for publication in Proceedings B. Congratulations!!

Open Access

Paper charges

Sincerely,

Dr John Hutchinson

Associate Editor:

Board Member

Comments to Author:

The authors have performed simulations to assess whether the imbalance between rare and abundant species had an impact on the model results. I am happy with their response to the continued reviewer concerns and addition to the manuscript, and I thank the authors for their time in shoring up their conclusions and honing an already-excellent manuscript.

Appendix A

Review of MS# RSPB-2019-2099: "Redundancy buffers functional diversity against extinction in Caribbean mollusks" by Christine Bacon, et al.

Reviewer: Phil Novack-Gottshall (pnovack-gottshall@ben.edu, Benedictine University, Lisle, IL)

Overview

This manuscript has many strengths. The foundation is a comprehensive (and scientifically invaluable) database of functional traits for >5000 species (essentially all) of Miocene–Pleistocene Caribbean gastropods and bivalves. The data are used to evaluate diversification (especially extinction) dynamics during this critical interval in earth history, when oceanographic changes in the Panamanian isthmus fundamentally altered the marine ecosystem in this basin, and throughout the world. The authors conclude that taxonomic and functional diversity (measured using multiple complementary metrics) increased gradually through the Miocene, reached peak diversity in the Pliocene, then suffered rather substantial declines during the Pleistocene glacial perturbations. Most importantly, they demonstrate that despite major species extinction (on the order of 30%) during the crisis, functional diversity was fundamentally unaltered because most extinctions affected species that had functional complements somewhere else within the basin. They conclude that these patterns point to the importance of competition in structuring the functional space during environmental crises. These are significant conclusions, and one likely to be widely cited by both paleontologists, ecologists, and conservation biologists.

The manuscript has other analytical strengths, including the addition of a multi-trait-dependent extinction (MTE) model to one author's PyRate Python package, which allows relative weighting of which functional traits were especially correlated with likelihood of extinction. These analyses allow the authors to conclude that life habits with high redundancy (that is, in which many species shared essentially similar niches) were most likely to go extinct, not only during the Pleistocene crisis but also in background times throughout the Neogene. Body size, and to a lesser extent diet and environments, also played important roles during individual epochs. Analyses that can integrate such extinction risks within broader diversity analyses are especially welcome for future studies.

I am not an authority in the Bayesian analyses used herein, so cannot confirm whether the analytical decisions (parameterization of priors, details for MCMC trials, etc.) are appropriate and analyzed correctly. Given the ample size of the data set, I suspect the results would be robust regardless of the statistical method employed. Possible suitable reviewers to consider to review such particulars include Bjarte Hannisdal, Steve Wang, David Bapst, and Lee Hsiang Liow.

My overall view is that this manuscript ought to be accepted for publication. However, I have one concern, and several minor ones, that I think the authors ought to address.

Major concern #1: Statistical tests should be included to evaluate interdependencies in the trends in fig. 2.

The patterns reported in figure 2 summarize visually the major results of their primary analysis. Accordingly, the authors draw numerous evolutionary and ecological conclusions based on their interpretations of how these trends diverge from each other. (For example, in lines 166–169 they conclude

the Neogene diversification occurred via "overall expansion" of ecospace, based on the increasing trends in functional entity richness in fig. 2e in the face of a "steady" and unchanging over-redundancy in fig. 2h). However, both trend lines appear to be largely increasing to me. (And with fig. 2h potentially not plotted in the same manner as the other statistics, see below, the increase is visually more dampened than it likely really is.)

Without a statistical correlation analysis (using first-differences to detrend the time-series), I am not convinced these dynamics are acting independently from each other. (In addition to, or in place of, this, it might be worthwhile to break the tests into epoch-level tests, as it seems clear the dynamics may have changed through time; similar dynamics during one long epoch like the Miocene should not dilute those epochs when they may have differed, such as during the Plio-Pleistocene.) In fact, the functional and taxonomic trends in Fig. 2 all appear quite correlated to me, which could point to different diversification dynamics than interpreted by the authors.

Within this context, note that I define "ecological expansion" in a potentially different manner than the authors do. I consider expansion as a pattern in which new functional entities are more extreme than previously occurring, typically observed by increasing functional richness without an increase in redundancy (which they demonstrate does increase at the same time). In contrast, a more trait-independent diversification (which I have terms a neutral or null model of diversification), in which trait identity has no bearing on diversification dynamics and newly originating species simply evolve new traits at random, would also result in some of the patterns you report here. Namely, an increasing FE richness, richness, and redundancy. (See Bush and Novack-Gottshall [2012, Modelling the ecological-functional diversification of marine Metazoa on geological time scales. *Biology Letters* 8:151-155] and the more detailed follow-up of Novack-Gottshall [2016, General models of ecological diversification. I. Conceptual synthesis. *Paleobiology* 42:185-208] for additional details on how such statistics can change during ecological diversifications.)

In my opinion, the diversification trends are overall in line with what one would expect during random exploration of ecospace, rather than a competition-driven milieu, as the authors suggest. I am fine with their take on the causes, assuming the correlation among patterns are truly as they describe. Statistical tests can confirm one way or another. Tightening language on what the ultimate ecological processes that structured these ecosystems might be another strategy.

(Along these lines, why not include an ordination of the functional trait space through time, as was done in Pimiento, et al. (2017)? Inclusion of these graphs ought to be straightforward to obtain, given the data and analyses, and could substantially bolster understanding the processes that drove the dynamics herein. Not required, as the manuscript is quite self-contained, but if you want to further develop the processes—competition, increasing specialization, exploration of ecological novelty, or just plain "neutral" stochastic processes—then such ordinations would be helpful.)

Minor concerns worth considering

Saturation?—The authors' claim of "saturation" (in line 169) is not warranted. It seems you are assuming the interval of "maximum observed functional diversity" is the same as "saturated," but it is possible

(even likely) that additional functional entities could have yet evolved (i.e., the ecosystem could have accommodated more life habits) if the Pleistocene glaciation had not occurred. With the trait-space you use, it is possible for $6 * 6 * 6 * 7 * 7 = 10,584$ unique functional entities to exist. You note (line 313) there were a possible 7,776 ones ($=6^6$), obtained by removing pelagic and freely-fast locomotion and restricting analyses to benthic mollusks in the process. All good. You note (line 315) your data set includes only 369 functionally unique entities (and it could include 1,899 if treating each body size as a separate trait). Either way, this is a very small number of the possible number of functional entities that could have existed. So I remain skeptical that the functional space was truly "saturated" in these communities. Additional analyses could convince me otherwise.

Functional traits—What is a "detrital" feeder? Also, you note (line 291) your categories are based on those of Bambach, et al. (2007), yet their categories are not the same as you use. For example, they did not use body size, which is an important addition in your data set. Nor environment. Consider further noting the ways you modified their categories, defining or justifying new traits and categories. (Note this recommendation should not be taken as a criticism! I am pleased with the traits you used.)

Comments on tables, figures, and references

Fig. 1: This figure is acceptable, but it's not easy to view the record coverage without zooming. Perhaps the dark gray map outline can be lightened? Or the record symbols be made slightly larger? I recognize there's a trade-off between seeing each record separately and making them large enough, so it's possible this graph is at the optimum. I think, instead, it's close but not quite there. If possible to make it easier to read, it would be worthwhile to alter the figure slightly.

Fig. 2: Explain in caption how the confidence regions were calculated.

Fig. 2h: It's not clear why this statistic, although described as percent-maximum-transformed, does not surpass a value of ~ 0.5 . Shouldn't it span a greater range of values, as the other percent-max-transformed statistics do? If not, perhaps explain why in corresponding text.

Fig. 3: The use of colors green, orange, and red for the "log Bayes factor" levels can cause confusion to readers, as they are the same colors used to designate time intervals. Change to a different (and unique) color or pattern.

S1: Why no deposit feeders? Nuculoids, nuculanoids, and some tellinoids and lucinoids ought to be present, no? Perhaps these are coded as detritus feeders? If so, please clarify terms. Later in fig. S6, these appear to be present, so perhaps fig. S1 needs to be replotted to include this category?

Fig. S6: Not clear why this figure exists, as it is re-plotted in Fig. 3 (bottom left).

Table S1: ">" in caption title is mis-leading, as implies you only counted values "greater" than the null predicts. Ought it be "more extreme than" or "> or <" instead of ">"? Please clarify.

Table S2: The rows are not clearly aligned. Assuming "ordered factor" refers to Environment and not Locomotion?

Should be cited:

Discusses functional impact of P/Tr and K/T on bivalve extinctions, showing maintenance of high functional redundancy across these events: Edie, S. M., D. Jablonski, and J. W. Valentine. 2018. Contrasting responses of functional diversity to major losses in taxonomic diversity. *Proceedings of the National Academy of Sciences* 115(4):732-737.

Minor (line-item) comments (by line number):

Line 55: Consider citing Edie, et al., 2018 (citation above) here and/or elsewhere.

73–74 and 91–93: The introductory sentences are too redundant. Consider cutting the first usage (lines 73–74) and moving Jablonski reference here.

99: change "Addressing" to "address"

101–103: Sentence is not grammatically parallel: third clause is not independent as the first two are.

149: "function entity" should be "functional entity"

150: additional clarification on what "volume" "functional richness" is measuring is needed here. Is this a PCoA or other ordination convex hull hypervolume? Something else?

152 & 156: Optional: "Notably" and "Importantly" are not really necessary; better to let the results speak for themselves.

169: "figure 2G" should be lowercase "figure 2g".

190: End of sentence is grammatically incomplete; consider inserting "those that were functionally unique".

208: Insert "to" to "due to Quarternary"

273: "time variable preservation" should be hyphenated "time-variable preservation"

434: should it be "functionally similar" or "functionally identical" here? In other words, are you treating redundancy as functionally similar or functionally identical? (Lines 64 and 435 imply you are treating it in the conventional manner, of being functionally identical.)

Appendix B

Dear Professor Hutchinson,

We are grateful for the invitation to submit a revised version of our paper following its initial review.

Overall, the extensive comments provided by the reviewers allowed us to greatly improve our manuscript. As such, we have completely reanalyzed our data based on only tropical Caribbean data. Further, we have improved the explanation of our methods and expanded our discussion. Most critically, we have tested the role of preservation and occupancy on our extinction models, which enabled us to clarify the robustness of our results.

Below we provide a point-by-point response (in blue) to each comment.

We hope this improved version is now suitable for publication in *Proceedings of the Royal Society B*.

On behalf of all co-authors, thank you very much for considering our manuscript.

Thank you for your consideration,

Christine D. Bacon
Associate Professor

University of Gothenburg, Gothenburg, Sweden

Gothenburg Global Biodiversity Centre, Sweden

Universidad CES, Medellín, Colombia

christinedbacon@gmail.com

Associate Editor

Board Member: 1

Comments to Author:

The authors use an impressive dataset and a sophisticated suite of analyses to address how functional diversity has changed over the last 23 million years using Western Atlantic molluscs. The reviewers and I generally find the study of value and interest to the broad readership of PRSB. The study aims of identifying extinction mechanisms are particularly timely given the current high rate of biodiversity loss.

We thank the editor for their commitment to the manuscript and the summary of our reviews.

However, a number of concerns were raised regarding the manuscript and accompanying analyses, which the authors should find useful when revising. All reviewers, for example, raised concerns about the clarity of the methods as presented, such that a revised version of the manuscript should ensure each analysis is reproducible. All reviewers also questioned how spatial coverage and sampling could affect patterns obtained, and I encourage the authors to consider sensitivity analyses to address these concerns.

We have improved the Methods and assessed the role of spatial coverage (occupancy) and preservation biases in our extinction models. We are very thankful for these comments, as our tests allowed us to further clarify the robustness of our results.

Reviewer 1 mentioned that the Discussion needs expansion, and that the study aims/questions (Intro) need to be reformulated to reflect the nature of the analyses conducted. Reviewer 2 raised a concern regarding the lack of statistical tests to analyse the major patterns reported in Figure 2, which I echo, and questioned some of the patterns the authors suggest based on the data presented. The authors should consider whether the diversification trends truly reflect competition or could simple result from a random exploration of ecospace. Reviewer 3 mentioned that the pulse of extinction cited at 3.2 to 2.6 Ma does not correspond with the data presented, which instead indicates a major pulse of extinction at 3.5 to 4 Ma. Both Reviewer 3 and I also question the inclusion of areas such as North Carolina to Florida in a definition of ‘Caribbean’, and some justification of this choice is warranted. Finally, Reviewer 4 wondered about the role of abundance as a possible confounding factor in the association between functional traits and extinction.

We have significantly expanded the Discussion and reformulated our research questions in the Introduction. Further, we have clarified the timing of the extinction phase throughout the manuscript. Importantly, we made great efforts to completely reanalyzed our data based on only tropical Caribbean data and including new analyses that address the concern of the role of abundance on the results from our extinction models.

These are not insignificant issues, and they will need to be addressed in order to be considered suitable for publication.

Referee: 1

Comments to the Author(s)

This is the report for «Redundancy buffers functional diversity against extinction in Caribbean mollusks» by Bacon et al. In this study, the authors analyze an impressive dataset of 44,000 mollusks fossil occurrences in the Caribbean region and couple it to a trait dataset using an

advanced Bayesian approach. The authors aim to study diversity changes from the Neogene to the Quaternary and assess whether species extinctions were linked to the species' traits (identity and redundancy). The authors find that extinctions were not occurring randomly across species but that extinction rates were higher for species that tended to be redundant (their trait characteristics are shared by other species). As a result, functional richness (calculated as the volume in the trait space, i.e. trait richness) stayed significantly high (higher than expected by chance) after extinction events. This suggests that ecosystem with high functional redundancy might be more resilient to extinction events.

The manuscript of Bacon et al. brings novel insights into the mechanism of species extinction and is extremely relevant under the ongoing climate crisis to assess which species are more likely at risk of extinction (i.e. redundant species). I enjoyed reading this high-quality manuscript and I believe it will be of wide interests to the readers of RSPB.

I have however some comments and suggestions for clarifications. I would especially like the authors to expand the discussion, which is for now minimal. Among other I would like them to discuss the potential limitation of the used traits (e.g. if they had selected other traits & categories or added more traits how could this have impacted the functional redundancy results?), and of the methods (fossilization rate, whether there is potential trait-based bias in encounter rate for e.g. smaller species). Discuss also why some trait categories are linked to a higher extinction rate as well as on the relevance of the results. I also have some comments on the method used for doing the null model (is the frequency of occurrence taken into account?) and the calculation of the species-level redundancy, but this might just be a misunderstanding due to some lack of precision in the M&M. Please find more details on these issues below.

We thank the reviewer for the constructive comments which we addressed in our revised manuscript, as detailed in our response below. We have significantly expanded the Discussion, particularly addressing why some trait categories are linked to a higher extinction rate as well as on the relevance of the results.

Regarding the use of traits, we agree in that there are limitations in data, and the number and type of traits used can profoundly affect functional diversity measures (Lefcheck *et al.* 2015; Zhu *et al.* 2017). Here, we leverage five traits that are commonly used to measure the ecology of mollusks. To address the reviewer's point we tested the effect of our trait selection on our results by repeating our functional redundancy analyses dropping one and two traits at a time (Supplementary Methods). Altering the number of traits did not change the pattern observed in functional redundancy (figure S7).

The reviewer is correct in pointing out the potential issue of fossilization bias in traits. In response, we performed a new test by looking at the number of occurrences per species as a proxy for preservation potential and plotted this proxy by trait state (Supplementary Methods). We found limited differences in number of occurrences per species across traits and states (figure S6). Therefore, we consider this potential issue not having a major effect on our results.

Introduction:

L101-103: I don't think the questions formulated here really reflect the main analyses of the

study, please reformulate. Question 1 & 2 appear to be the same (functional diversity included in “diversity”) and 3 not targeted particular “species” but particular “traits.

We agree and have rephrased this section in the introduction.

M&M:

Is the fossilization rate the same between the different mollusks class? Can some of the studied traits have some influence on the fossilization or fossil encounter rate? And what could be the impact of this potential bias on the results? If I understood correctly, the authors did take into account the potential bias in their Bayesian approach, but it would be good to have this potential limitation stated much more explicitly to build more trust on the results.

In our analyses we incorporate temporal variation in preservation rates but assume homogeneous rates across taxa. Although models to allow rate heterogeneity across taxa exist, we could not implement them here given the large size of the dataset, as they would have been computationally prohibitive. However, the test for trait-associated preservation biases, described in our response above, did not point to a high heterogeneity of preservation rates among the taxa in our dataset (figure S6).

L251-253: this is not really clear. So each record dated to a specific geological epoch was assigned as being present during all of this epoch? While in L270 the authors mention “the observed temporal range between first and last appearances”. Understanding how the different species were assigned as present/absent in the different time bin (1Myr) is crucial to understand well the method, so I will urge the authors to present more clearly the different sequential steps that were required to obtain the species*time occurrence matrix (L298).

We have simplified the section to clarify this issue (section a in Methods). Essentially, each occurrence collected had an age range associated with it. We refined this range using published references to place them into the finest stratigraphic resolution possible. For example, if an occurrence was found to be Miocene from the Gatun Formation in Panama, we assigned it to *late* Miocene based on the current literature on the geology of this area. Later, as part of the PyRate analyses (section b in Methods), times of origination and extinction were estimated (simultaneously with speciation and extinction rates per time unit, which in our case is 1 Myr). This part of our analyses effectively infers the longevity of each species (e.g., species *x* is extant and lived from 12 to 0 Ma). This range is later used in the functional diversity analyses (section c in Methods) where the presence/absence of each species is binned across the last 23 Ma (e.g., species *x* is absent in time bins from 23 to 12 Ma, and present in the time bins from 12 to 0 Ma).

L266. Please explain more what the preservation biases refers to and how it is taken into account. Preservation biases refer to incomplete sampling due to fossilization and preservation processes. We have now rephrased this in the manuscript (lines 344-349 and 355-356).

L290: state the 5 traits. What is the criteria for differentiating deposit feeders and detrital?

Deposit feeders (and suspension feeders) could feed on detritus?

The now list the five traits (section c in Methods) and the character states are found in table S1, figure S6 and in the electronic supplementary dataset.

The differentiation derives from where those bivalves source detrital material- buried sediment (=mining) or surficial sediment (=surface deposit feeding). There tends to be major anatomical

differences between those infaunal bivalves consuming organic material buried in the sediment and those that do so from the surface. A clear example of this is the presence of inhalant siphons, and indeed suspension-feeders. Both live in the same environments, but are accessing organic nutrients in distinct ways, and in ways that allow them to exploit spatial and temporal variations in the quality and quantity of food, possibly make them more or less vulnerable to predation, achieve varying body size, probably reproductive strategies, etc. Some surficial deposit feeders may facultatively suspension feed (e.g., some of the larger Tellinidae) and these were coded as such. Some filter feeds may well ingest deposited material too. Nevertheless, the various groups reflect distinct feeding strategies and different food sources with distinct morphologies.

L290: rephrase. “structure” is often used to refer to species identity, as you are using traits here it might be better to choose word such as “characterize the ecological niche” (and delete the last part of the sentence “functional diversity”).

Done.

L319: the implementation of the null model needs more details. The null model was done per time bin and not across time bin? If across time bin, did you also take into account the different species frequencies when doing the randomization? When doing community null model the randomization should not only keep species richness constant (keeping the same number of species for each time bin and randomly selecting species from the species pool) but also taking into account the frequency of occurrence of species (species that are occurring in many time bins are more likely to be randomly selected than species in few time bins).

In our randomizations we shuffled the species identities, while maintaining the pattern of diversity changes constant over time. We did not resample species proportionally to their lifespan, as the aim of the simulations was only to address whether any random drop in species diversity due to increased extinction would predict a change in functional diversity similar to that observed in our data. We have expanded our explanation of the null model in a the Supplementary methods (section Comparative simulations for functional diversity in the Supplementary methods).

L441: Is the species-level functional redundancy estimated using the entire species pool (irrespective of time)? A species might be redundant when aggregated for the complete studied period but be unique during a time-bin? To be in line with the question addressed here, it might be better to calculate species-level functional redundancy as the average functional redundancy of the species across time-bins, although I do not think this would change much the results. Functional redundancy, and all functional diversity indices were computed per time bin, relative to the overall assemblage. However, we also report the mean level of redundancy of the whole assemblage in the methods section. We have clarified this throughout the manuscript.

Results & Discussion:

L278: I didn't understand all the resort of this analysis, but from the Fig S5 it seems high only from -5Ma?

The reviewer is correct. We now clarified that preservation rates are comparable to (or higher than) preservation rates used in previous simulation studies, which demonstrated the robustness of the method applied here (lines 363-365). Thus, we expect our results to be robust.

I would like to see the discussion expanded especially on the traits identity that appears to have high effect on extinction rates, and on how these results can inform on the currently threaten ecosystems e.g. resilience and management (building up on what is stated in L99 “could potentially help addressing current and future extinction scenarios).

We have expanded our discussion based on new results, but we have removed our original, introductory statement that the results could give perspective on community response to current and future scenarios.

L204: This is somehow an unexpected result, as large species are often considered more vulnerable and thus prone to extinctions. If I understood correctly, the argument of the authors is that this might be because there is a higher number of small species (higher redundancy). I am wondering whether it would not also be interesting to standardize the extinctions per number of species in each trait category i.e. proportionally would the extinction rate per size class then changes and become higher for larger species? I think it is important to make a distinction between absolute/relative extinction and I would at least like to have more discussion on this size-based extinction rate as it is not the results one would especially expect.

Large species have been found to be consistently more vulnerable during the current extinction crisis (Dirzo *et al.* 2014 Science; Ripple *et al.* 2017 PNAS). In the geological time; however, results vary with some studies finding the small organisms are more prone to extinction and others that large organisms are (Harnik *et al.* 2012 TREE). We have included this in the new version of the manuscript. Nevertheless, our argument stems from the fact that ecosystems tend to hold more small species than large (e.g. Kozłowski & Gawelczyk, 2002 Func. Ecol.). Thus, among small-sized organisms, there is likely to be more scope for functional insurance provided by redundant species (see Duffy, 2003 Ecol. Lett. and Duffy, 2002 Oikos). The reviewer makes an interesting suggestion to standardize extinctions per number of species in each trait category; however, as shown in our new figure S6 (also see our response to first concern) we found limited differences in number of occurrences per species across traits and states.

L181. In Fig 3 (and Fig S6) “suspension” feeder species does not appear to have an especially high extinction rate, remove. In the introduction, L82 you state that other studied demonstrated a higher extinction rate for suspension feeding bivalves and predatory gastropods, which you do not find here. What could be a reason for the observed difference? Please discuss this result more. Also, what is the difference between figure S6 and 3?

We have largely modified our figures in light of the new results. There is no repetition in the figures in this improved version. Indeed, and in contrast to previous studies, neither suspension feeding or predatory gastropods had higher extinction rates, but those at depths between 15-20 m did, which we take together to suggest that species living in such environments would be more affected by sea level changes, area loss and consequent extinction.

L225: think of rephrasing, as you do not explicitly analyzed the temporal changes in diversity vs environmental change (e.g. “coincide with” instead of “was related”).

Done.

Others:

The key word mention “Isthmus of Panama”. It is not referred to anywhere else in the text.

Consider changing the location mention.
We removed this keyword.

In Fig S1. Deposit is missing from the diet histogram.
This figure has been replaced. The new figure includes all trait categories (figure S6).

Fig S2. Line 3, what does CI 10-14 refers to?
Credible intervals. This has been clarified.

Table S2. “chemo or photo symbiotic” diet are not on the histogram fig S1
Chemo or photo-symbiotic are the specialized diet type. We have fixed this (now table S1).

Referee: 2

Comments to the Author(s)

This paper analyses a large dataset for extinction patterns in taxonomic and functional diversity in West Atlantic mollusks during the Neogene. Like many authors, they find an extinction pulse in the late Neogene, and as reported at several other, larger extinction events, see modest loss of functional diversity relative to taxonomic losses at this time.

The ms states that “over the last three million years, Caribbean mollusks went through a period of high extinction driven by oceanographic changes.” This is not strictly equivalent to the statement in the text that the maximum extinction is “between 3.2 and 2.6 Ma (figure 2b)”. More importantly it does not match the figure, which clearly shows the major pulse in extinction rates to be in the Early Pliocene, from 3.5 or 4.0 Myr ago to a little more than 5 Myr (in binned terms, the Zanclean, 3.6-5.3 Ma). This is not a trivial issue, as the explanation for the observed patterns hinges in part on the timing of geological events. They further note the suggestion “that the extinction of mollusks presented a million-year temporal lag between environmental change and detectable extinction, becoming extinct between 2 and 1 Ma,” an even larger discrepancy with their data; this contradiction is also never addressed.

We re-run our analyses using samples from the Caribbean only (see comment below), so these results have changed. We have now clarified times of extinction and the differences between these and shift rates. That said, please note that peaks in extinction, or maximum extinction refers to the times where the extinction rates were the highest. Accordingly, it does not refer to where extinction rates *start* to increase.

Regarding the lag, please note that such statement is related to a previous study and it is only mentioned in the introduction (reference provided). Therefore, it does not represent a discrepancy in our data. Nevertheless, in light of the reviewer comment, in this improved version we discuss this in the Results and Discussion section.

It has long been known (for example Foote 2003, 2010, but many other sources) that artificial origination pulses occur in intervals following times of poorer sampling (just as artificial extinction pulses precede times of poor sampling). This sampling effect could explain the supposed high rate of origination in the Aquitanian relative to later intervals, given that the preceding Chattian interval has ~430 occurrences (vs 660 occurrences in the following

Aquitanian interval and ~5000 occurrences in the Burdigalian interval after that); evidently PyRate is sensitive to some types variations in sampling/preservation but not others, and clearly this point needs to be clarified; perhaps it accounts for the different results obtained by Crouch & Clarke 2019 when analyzing data using PyRate and the TRiPS method of Starfelt & Liow 2016 (Phil Trans).

We accounted for sampling heterogeneity through time in our analyses, since PyRate jointly models speciation, extinction, and preservation. The Chattian was not included in the analyses, which were restricted to the Neogene and Quaternary, as we now clarify in the manuscript (e.g. section a of the Methods). We emphasize the preservation rates were well within the range of values PyRate is known to work well with (based on previous simulation studies). We now discuss this on lines 359-361.

Finally, we also note that the apparent differences between PyRate and TRiPS described by Crouch & Clarke (2019) are (to the best of our understanding) the product of a misinterpretation of sampling “rates” (PyRate) vs. sampling “probabilities” (TRiPS). A rate can be translated into a probability per time unit using the formula:

$$\text{Prob} = 1 - \exp(-\text{rate})$$

Applying this formula to the results of Crouch & Clarke (2019) is likely to show that the probabilities estimated under the two methods are in fact very similar.

In a related point, a clear explanation is also needed for the estimate of constant preservation rates through time from ~23 Ma to ~5 Ma, given that the number of occurrences per bin within that interval vary from 662 to 5500 or 6000, depending on how composite time bins are treated. How, in fact, were occurrences in composite time bins treated (Burdigalian-Langhian, etc.)? Was every occurrence counted once in each of the two time bins, were they apportioned in some way between the two time bins, were they counted in the continuous time series as intermediate in age between the two bins, or as present in every one-Myr increment through the composite bin? The number of occurrences in such bins ranges from ~160 to more than 6700, so this is not a trivial analytical decision, but it is not referred to in the text or supplement.

A number of results rely on conversion of the time-binned data to continuous timeseries. For example, the authors mention a low-origination interval extending 5-10 Ma, followed by an origination pulse at 4-3.4 Ma. How was this pulse constrained so tightly when the Early Pliocene data occur in a bin dated at 3.6-5.3 Ma? Similarly, the ms needs to clarify how the speciation pulse shown as a plateau from 20-22 Ma was generated when the raw data fall within the Aquitanian stage, which starts at 23.7 Ma, when there is a trough shown in the speciation rates. These issues probably point to the need to explain how they converted their time-binned data to continuous timeseries as shown in the figures, given the potential for biases introduced by time-binned data, particularly with unequal time bins as used here (ranging from 0.9 Ma to 6.6 Ma in length) (see Foote 2000, Alroy 2010, and many other references). Were the biases tested for, or otherwise shown to be minor effects? The paper mentions “time windows”, are these equal to the stage-level bins tabulated in the Supplement, or are these 1-Myr increments, as implied in Supplement Tables S1 and S3.

We now explain in more detail the procedure used here (section b in the Methods), which is based on previous studies by Silvestro et al. 2014 Syst Biol, and 2019 Paleobiology. All PyRate

analyses were run in continuous time based on fossil ages resampled within their stratigraphic range 10 times and then combined to incorporate dating uncertainty in the results.

In the .csv dataset, Column G gives the geo.region for all localities, including those whose latitudes correspond to South Carolina and Georgia, as “Caribbean”. Authors such as Woodring, Petuch, and Vermeij have shown that North Carolina-Florida was a distinct biogeographic province through most or all of the study interval, with a different fauna and spatio-temporal dynamics from the core Caribbean fauna (sometimes termed the Caloosahatchian and Gatunian provinces respectively). Some demonstration is needed that merging these two provinces in the present study is analytically appropriate.

We re-ran our analyses using only Caribbean occurrences.

The ms concludes that “molluscan communities experienced major diversity changes during the last 23 million years” but given that the analyses lump the entire region from Venezuela and Panama to Texas and South Carolina, no direct conclusions can be made at the community level. Although extinction did likely reduce alpha-diversity, no account is taken of changes in beta diversity, which might also contribute to net regional diversity increases and decreases, or as noted above, there no consideration of different diversity changes in the two provinces lumped together here.

Our reanalysis used only Caribbean occurrences.

It is difficult to evaluate spatial pattern in the data. Figure 1 is not very informative in that it does not indicate how the occurrences are distributed through time. It is striking how many of these points are in cool colors, indicating that they represent only 1~10 occurrences. Does spatial coverage, defined by locations yielding some minimum number of occurrences, change significantly through the timeseries? If so, how was that taken into account? The ms attributable increasing taxonomic and functional diversity to the increase in available habitat types, but all of their habitat categories are present in all of their time bins, and there is no treatment of the alternative sampling hypothesis, i.e. amount of each habitat type sampled in a time bin. Quantifying spatial coverage per time bin is important here.

The previous figure 1 was not very informative, we agree, we updated it and moved it to electronic supplement (figure S1).

Regarding the conclusions, Edie et al. 2018 PNAS, surprisingly not cited here, also found a mismatch between taxonomic and functional extinctions, at two major extinction events, the end-Permian and end-Cretaceous; given these and other prior results (Erwin et al. 1987 is the pioneering study that also found minor functional losses despite major taxonomic extinction, and also is not cited), is it surprising that a study of a smaller, regional extinction produced similar findings? Further, what is the actual mechanism for “selectivity against functionally redundant species”? I see that Edie et al. also discussed extinctions in these terms, and concentrated in the most richly occupied functional groups -- that paper certainly should be cited and the results here related to that study. Edie et al. outlined four hypotheses for the uneven extinction they also observed among functional groups, are any of those definitively rejected or supported by the analyses in the current ms?

We agree and have added the Edie et al. (2018) paper to our introduction and discussion.

Minor points

On Fig. 1, the data points should be larger. And it is not clear what the final phrase “and recent occurrence records are extant collections” is referring to.

We prefer to keep the point size as it is so we can more clearly map the number of records per site. We fixed the phrasing for readability. Please note this figure has been moved to the supplement.

It is difficult to navigate this .csv file, and an additional field should be added for placenames – minimally countries, and states for the larger countries such as Venezuela and the U.S.

We have modified the dataset to make it easier to navigate. We included additional columns to provide more information.

The min-age for the Chattian is given as 23 Ma, but the max-age for the immediately overlying Aquitanian is given as 23.7 Ma, yielding an overlap of the two timebins. Did this distort numbers in the 23 Ma time bin? The time series in Figure 2 shows only the Miocene-Recent, where do the 431 Chattian records enter into the analyses?

Even though we had data from the Chattian onwards, we focused on the Neogene and Quaternary, which we now clarify in the manuscript. Accordingly, all Oligocene records were excluded in our analyses by filtering the data in R. We left the Chattian records in the master dataset, but have now removed them to avoid confusion, following the reviewer comments. Also, please note there is no longer overlap in time bins.

What does “NA” in column A signify for the analyses? Were these 13,851 occurrences of 235 species omitted, or otherwise used differently from the occurrences that have numbers in column A? Were they sourced differently?

The new column A (id_datasource) lists the source where the data was collected. Those collections without a number (NAs) were a product of a file reading error which has been corrected. Accordingly, there are no NAs in the new version of our dataset.

Referee: 3

Review of MS# RSPB-2019-2099: "Redundancy buffers functional diversity against extinction in Caribbean mollusks" by Christine Bacon, et al.

Reviewer: Phil Novack-Gottshall (pnovack-gottshall@ben.edu, Benedictine University, Lisle, IL)

Overview

This manuscript has many strengths. The foundation is a comprehensive (and scientifically invaluable) database of functional traits for >5000 species (essentially all) of Miocene–Pleistocene Caribbean gastropods and bivalves. The data are used to evaluate diversification (especially extinction) dynamics during this critical interval in earth history, when oceanographic changes in the Panamanian isthmus fundamentally altered the marine ecosystem in this basin, and throughout the world. The authors conclude that taxonomic and functional diversity (measured using multiple complementary metrics) increased gradually through the Miocene, reached peak diversity in the Pliocene, then suffered rather substantial declines during the

Pleistocene glacial perturbations. Most importantly, they demonstrate that despite major species extinction (on the order of 30%) during the crisis, functional diversity was fundamentally unaltered because most extinctions affected species that had functional complements somewhere else within the basin. They conclude that these patterns point to the importance of competition in structuring the functional space during environmental crises. These are significant conclusions, and one likely to be widely cited by both paleontologists, ecologists, and conservation biologists. The manuscript has other analytical strengths, including the addition of a multi-trait-dependent extinction (MTE) model to one author's PyRate Python package, which allows relative weighting of which functional traits were especially correlated with likelihood of extinction. These analyses allow the authors to conclude that life habits with high redundancy (that is, in which many species shared essentially similar niches) were most likely to go extinct, not only during the Pleistocene crisis but also in background times throughout the Neogene. Body size, and to a lesser extent diet and environments, also played important roles during individual epochs. Analyses that can integrate such extinction risks within broader diversity analyses are especially welcome for future studies.

I am not an authority in the Bayesian analyses used herein, so cannot confirm whether the analytical decisions (parameterization of priors, details for MCMC trials, etc.) are appropriate and analyzed correctly. Given the ample size of the data set, I suspect the results would be robust regardless of the statistical method employed. Possible suitable reviewers to consider to review such particulars include Bjarte Hannisdal, Steve Wang, David Bapst, and Lee Hsiang Liow. My overall view is that this manuscript ought to be accepted for publication. However, I have one concern, and several minor ones, that I think the authors ought to address.

We thank the review for their constructive feedback and we have gone to great lengths to address each concern and hence, present a much-improved revision of our work.

Major concern #1: Statistical tests should be included to evaluate interdependencies in the trends in fig. 2.

The patterns reported in figure 2 summarize visually the major results of their primary analysis. Accordingly, the authors draw numerous evolutionary and ecological conclusions based on their interpretations of how these trends diverge from each other. (For example, in lines 166–169 they conclude the Neogene diversification occurred via "overall expansion" of ecospace, based on the increasing trends in functional entity richness in fig. 2e in the face of a "steady" and unchanging over-redundancy in fig. 2h). However, both trend lines appear to be largely increasing to me. (And with fig. 2h potentially not plotted in the same manner as the other statistics, see below, the increase is visually more dampened than it likely really is.)

We have re-ran our analyses using a more refined geographic range (see responses above). As such, our plots have slightly changed. Nevertheless, please note that we plotted all our functional diversity figures in the same manner using the same time bins. In terms of the increase in functional diversity, to clarify our point, we have only left in figure 2 two functional plots: e) functional redundancy and f) functional richness. In the text, we specifically refer to these indices (and plots) when we describe the general increase, which we think is clear during most of the Neogene. We specifically calculated this increase and reported in the results (68% increase in functional redundancy and 31% in functional richness). This increase is also evident in terms of the number of functional entities (58% increase; figure S4). Although functional over-redundancy also shows a slight increase, it is much lower than the other indices during the

Neogene (18%). In light of the reviewer's comment, we have modified our wording and now refer to a low increase in over-redundancy instead of "steady".

Without a statistical correlation analysis (using first-differences to detrend the time-series), I am not convinced these dynamics are acting independently from each other. (In addition to, or in place of, this, it might be worthwhile to break the tests into epoch-level tests, as it seems clear the dynamics may have changed through time; similar dynamics during one long epoch like the Miocene should not dilute those epochs when they may have differed, such as during the Plio-Pleistocene.) In fact, the functional and taxonomic trends in Fig. 2 all appear quite correlated to me, which could point to different diversification dynamics than interpreted by the authors. Please note that all our functional diversity analyses were performed per time bin (also see our responses to reviewer 1) and relative to the overall community. As such, we calculated our indices at each time bin and presented the proportional value relative to the entire community during the entire time range studied. For example, functional richness was calculated as the percentage of the overall volume of the functional space at time t . Therefore, we effectively performed these analyses accounting for the dynamics of each time period (in our case, 1 myr intervals). Further, we argue that even though our diversification dynamic metrics may be correlated at some extent (e.g., as you increase species you increase functional space and redundancy the more species the more functions), communities can, and do show different relationships between these metrics, reflecting differences in the distribution of species in trait space and among functional entities. Therefore, taxonomic and functional trends can be decoupled as we show here for the case of functional richness (while diversity drops in the quaternary, functional richness remains unchanged). We have clarified this in the manuscript.

Within this context, note that I define "ecological expansion" in a potentially different manner than the authors do. I consider expansion as a pattern in which new functional entities are more extreme than previously occurring, typically observed by increasing functional richness without an increase in redundancy (which they demonstrate does increase at the same time). In contrast, a more trait-independent diversification (which I have termed a neutral or null model of diversification), in which trait identity has no bearing on diversification dynamics and newly originating species simply evolve new traits at random, would also result in some of the patterns you report here. Namely, an increasing FE richness, richness, and redundancy. (See Bush and Novack-Gottshall [2012, Modelling the ecological-functional diversification of marine Metazoa on geological time scales. *Biology Letters* 8:151-155] and the more detailed follow-up of Novack-Gottshall [2016, General models of ecological diversification. I. Conceptual synthesis. *Paleobiology* 42:185-208] for additional details on how such statistics can change during ecological diversifications.)

We agree with the reviewer and have therefore removed our mention of expansion and used "increase" instead.

In my opinion, the diversification trends are overall in line with what one would expect during random exploration of ecospace, rather than a competition-driven milieu, as the authors suggest. I am fine with their take on the causes, assuming the correlation among patterns are truly as they describe. Statistical tests can confirm one way or another. Tightening language on what the ultimate ecological processes that structured these ecosystems might be another strategy.

1. We do not argue that competition drove the diversification trends. Instead, that competition

mediated the response of species to *extinctions* only, as our trait-dependent models only account of the effect of traits and redundancy on extinctions.

2. As the reviewer points out, most of the functional diversity values are what we would expect by chance for most of the Neogene (figure S3; table S2). In light of the reviewer's comment we now discuss this finding.

3. We have changed our language regarding competition as mediator of extinction and explicitly pose this more as a hypothesis.

(Along these lines, why not include an ordination of the functional trait space through time, as was done in Pimiento, et al. (2017)? Inclusion of these graphs ought to be straightforward to obtain, given the data and analyses, and could substantially bolster understanding the processes that drove the dynamics herein. Not required, as the manuscript is quite self-contained, but if you want to further develop the processes— competition, increasing specialization, exploration of ecological novelty, or just plain "neutral" stochastic processes—then such ordinations would be helpful.)

All our functional diversity analyses follow the methods of used in Pimiento et al. 2017. We have now clarified this in the manuscript. We considered the suggestion of including plots of the functional space, but they do not add new information to our manuscript.

Minor concerns worth considering

Saturation?—The authors' claim of "saturation" (in line 169) is not warranted. It seems you are assuming the interval of "maximum observed functional diversity" is the same as "saturated," but it is possible (even likely) that additional functional entities could have yet evolved (i.e., the ecosystem could have accommodated more life habits) if the Pleistocene glaciation had not occurred.

We agree with the reviewer. We have changed the word saturation to increase.

With the trait-space you use, it is possible for $6 * 6 * 6 * 7 * 7 = 10,584$ unique functional entities to exist. You note (line 313) there were a possible 7,776 ones (=66), obtained by removing pelagic and freely-fast locomotion and restricting analyses to benthic mollusks in the process. All good. You note (line 315) your data set includes only 369 functionally unique entities (and it could include 1,899 if treating each body size as a separate trait). Either way, this is a very small number of the possible number of functional entities that could have existed. So I remain skeptical that the functional space was truly "saturated" in these communities. Additional analyses could convince me otherwise.

We realized there was a mistake in table S1 that suggested there were seven diet categories in some traits when there are actually six. We have corrected this (see response referee 1 and table S1). Accordingly, we have $6 * 6 * 6 * 6 * 6 = 7,776$ FEs. From these, our data occupies 352 FEs in total (new dataset using a more restricted definition of Caribbean; see our response to referee 2). This is indeed a very low percentage of the possibilities (4.53%; also see Pimiento et al. 2017 and Mouillot et al. 2014 for examples with similar, low percentages).

Functional traits—What is a "detrital" feeder? Also, you note (line 291) your categories are based on those of Bambach, et al. (2007), yet their categories are not the same as you use. For example, they did not use body size, which is an important addition in your data set. Nor environment. Consider further noting the ways you modified their categories, defining or

justifying new traits and categories. (Note this recommendation should not be taken as a criticism! I am pleased with the traits you used.)

We define detrital feeding above in response to referee #1 and traits were based on Bush AM, Bambach RK, Daley G. 2007 Changes in theoretical ecospace utilization in marine fossil assemblages between the Mid-Paleozoic and Late Cretaceous. *Paleobiology* **33**, 76-97. We have clarified this in the manuscript.

Comments on tables, figures, and references

Fig. 1: This figure is acceptable, but it's not easy to view the record coverage without zooming. Perhaps the dark gray map outline can be lightened? Or the record symbols be made slightly larger? I recognize there's a trade-off between seeing each record separately and making them large enough, so it's possible this graph is at the optimum. I think, instead, it's close but not quite there. If possible to make it easier to read, it would be worthwhile to alter the figure slightly. We agree with the reviewer that the map figure is confusing and we have therefore moved the revised figure to electronic supplemental material.

Fig. 2: Explain in caption how the confidence regions were calculated.
Done.

Fig. 2h: It's not clear why this statistic, although described as percent-maximum-transformed, does not surpass a value of ~ 0.5 . Shouldn't it span a greater range of values, as the other percent-max-transformed statistics do? If not, perhaps explain why in corresponding text.

This plot has been moved to the supplementary materials. This is the level of over-redundancy. We have improved our explanation in the methods: percentage of species that fill FEs above the mean level of functional redundancy, that is, the overrepresentation of some FEs. As such, this index is rarely 1. Please note that the functional diversity indices (except for redundancy) are not transformed but provided as percentages because they are relative measures based on the total assemblage. Functional redundancy is the number of species per FE and was transformed to a proportion for consistency (but see figures S3-4).

Fig. 3: The use of colors green, orange, and red for the "log Bayes factor" levels can cause confusion to readers, as they are the same colors used to designate time intervals. Change to a different (and unique) color or pattern.

We have changed this figure considerably and the colors of the Bayes factors are now unique.

S1: Why no deposit feeders? Nuculoids, nuculanoids, and some tellinoids and lucinoids ought to be present, no? Perhaps these are coded as detritus feeders? If so, please clarify terms. Later in fig. S6, these appear to be present, so perhaps fig. S1 needs to be replotted to include this category?

There was a mistake with character states in feeding, but in the original submission. In our resubmission this information is found in figure S6 and deposit feeding is in the plot.

Fig. S6: Not clear why this figure exists, as it is re-plotted in Fig. 3 (bottom left).
Figure removed (but note that this is no longer applicable given the new results).

Table S1: ">" in caption title is mis-leading, as implies you only counted values "greater" than the null predicts. Ought it be "more extreme than" or "> or <" instead of ">"? Please clarify. We did count simply the frequency at which observed > null. Following standard two-tailed test threshold, if this frequency was greater than 0.975 or smaller than 0.025 we considered the observed significantly different from the null.

Table S2: The rows are not clearly aligned. Assuming "ordered factor" refers to Environment and not Locomotion?

We removed this column. All traits are ordered factors. We clarified this in the table legend.

Should be cited:

Discusses functional impact of P/Tr and K/T on bivalve extinctions, showing maintenance of high functional redundancy across these events: Edie, S. M., D. Jablonski, and J. W. Valentine. 2018. Contrasting responses of functional diversity to major losses in taxonomic diversity. *Proceedings of the National Academy of Sciences* 115(4):732-737.

Done.

Minor (line-item) comments (by line number):

Line 55: Consider citing Edie, et al., 2018 (citation above) here and/or elsewhere.

Done.

73–74 and 91–93: The introductory sentences are too redundant. Consider cutting the first usage (lines 73–74) and moving Jablonski reference here.

Done.

99: change "Addressing" to "address"

Done.

101–103: Sentence is not grammatically parallel: third clause is not independent as the first two are.

Done.

149: "function entity" should be "functional entity"

Done.

150: additional clarification on what "volume" "functional richness" is measuring is needed here. Is this a PCoA or other ordination convex hull hypervolume? Something else?

Done.

152 & 156: Optional: "Notably" and "Importantly" are not really necessary; better to let the results speak for themselves.

Done.

169: "figure 2G" should be lowercase "figure 2g".

Done.

190: End of sentence is grammatically incomplete; consider inserting "those that were functionally unique".

Done.

208: Insert "to" to "due to Quarternary"

No longer applicable.

273: "time variable preservation" should be hyphenated "time-variable preservation"

Done.

434: should it be "functionally similar" or "functionally identical" here? In other words, are you treating redundancy as functionally similar or functionally identical? (Lines 64 and 435 imply you are treating it in the conventional manner, of being functionally identical.)

No longer applicable.

Referee: 4

Comments to the Author(s)

The role of functional diversity has taken its rightful place in paleobiological studies, and this work by Bacon et al. takes it one step further by investigating the role of functional redundancy in extinction selectivity. This question has obvious implications not only for paleobiology, but also for conservation ecology and the ongoing mass extinction. The authors have compiled and augmented an impressive dataset on Caribbean mollusks spanning the last 23 million years, and they use a robust, probabilistic framework for estimating macroevolutionary rates and drawing inferences. Their results and conclusions come across as eminently sensible: functional diversity increases through time, and extinction rates are consistently higher in functionally redundant species, implying that selective extinction against functional redundancy enhances ecosystem resilience. Their results also suggest that competitive interactions mediate the effects of environmental drivers of extinction.

There is, however, one substantive issue that I believe the authors should address, and that is the role of abundance/rarity as a possible confounding factor in the association between functional traits and extinction. The trait distributions are presented in terms of richness per trait category, and one is left to wonder what those distributions look like in terms of occurrences (presumably, not vastly different, as there is no mention of standardized richness). In the case of the 'diet' trait, for example, the most frequent trait categories (e.g. predator, suspension) seem to have no relative effect on extinction (caveat: the deposit feeding category is missing from the histogram in Fig. S1). Is the relationship between a functional trait category and extinction decoupled from the abundance/rarity of species?

Addressing this question is important for two reasons:

(1) Confounding. Rarity may impart greater extinction vulnerability. If we allocate species to trait categories, and most species are rare (in a relative sense), then some categories may contain mostly or only rare species. If that were the case, then a strong association between the trait category and extinction may be due to rarity, rather than the trait itself. Conceivably, this

confounding could propagate into the functional redundancy-extinction association as well: if most species are rare, then we might see a kind of "functional central limit theorem" by which there are more ways to be rare than to be common and thus more opportunities for shared functional traits.

(2) Ecological implications. If rare species tend to be redundant and have little effect on ecosystem functioning, then their extinction may not be a cause for concern, it may even confer resilience. On the other hand, if the redundancy-extinction link is found to be decoupled from rarity, or if rare species tend to be functionally unique, they may instead buffer ecosystems (e.g. https://urldefense.proofpoint.com/v2/url?u=https-3A_doi.org_10.1371_journal.pbio.1001569&d=DwIFaQ&c=sJ6xIWYx-zLMB3EPkvcnVg&r=j5rzw3I5ZrkvSOJgwIecGCbp7BqlxpCZmRjd0zKgQ3c&m=FxtjXdGan5TCfsSSm1PfScKF1LOe12N5rco22f47BtU&s=i3PGNBEu0VEH4I5iVHMeRdm8eSHWsuThJTwXsc5H38&e=). Either way, such a finding in deep-time data would have rich implications for conservation.

Given fossil occurrence data, relative species abundance can be assessed using simple measures of regional occupancy (e.g. by gridding the occurrence data). With an occupancy-based measure of abundance (i.e. common and widespread, not necessarily locally abundant), it would be feasible to test the possible role of rarity in the relationship between functional traits and extinction. This test should not involve an unreasonable amount of additional work for this manuscript. The potential role of rarity in the redundancy-extinction relationship could possibly also be investigated by numerical experiment: given a log-normal species abundance distribution, assign species to arbitrary trait categories, and see if redundancy is associated with rarity, regardless of trait. Unless my suggestions can be dismissed on conceptual/logical grounds, I believe adding a rarity dimension would enrich the ecological interpretation and widen the implications of this study.

We thank the reviewer for these comments, which we have carefully considered, as shown below:

First of all, let us clarify that we have modified figure S1 (now figure S6) and now all trait categories are shown. Further, we now show occurrences per species for each trait category, as the reviewer suggests above. We found limited differences in number of occurrences per species (a proxy of abundance) across trait categories. This is true in the specific case of the two traits found to have *marginal* effects on extinctions: size and environment. For size, "small" was found to have some effect on extinction. This trait category is not more abundant, on average, than other size categories. For environment, shore-face was found to have some effect on extinction. This trait category is slightly more abundant than the other categories (new figure S6).

To assess the question of whether or not the relationship between extinction and traits is decoupled from the abundance/rarity of species we plotted abundance against the factors/traits found to have an effect on extinction: redundancy, environment and size (new figure S8). Abundance was assessed in two ways: 1) number of occurrences per species (also see figure S6) across the entire time interval studied (shown in the left hand panel of the figure below) and 2) occupancy as suggested by this reviewer, measured as the number of grids occupied per species,

where grids are defined by latitude and longitude cells (shown in the right hand panel of the figure below).

Redundancy: Our multi-trait dependent extinction model found high and very high redundancy levels had the strongest relative effect on extinctions (red boxplots above). Our analyses reveal that there are only small differences in occurrences across levels of redundancy. In terms of occupancy; however, we found ‘very high’ redundancy (the one with the highest effect on extinction, see figure 2b) to have mostly rare species. Nevertheless, the fact that the ‘high’ redundancy category (which was also found to have a strong effect on extinction) has occupancy no different to the lower redundancy groups, indicates that rarity does not bias our extinction selectivity results. In sum, our analyses show that the redundancy categories found to be selected against (= high and very high redundancy) have both rare and abundant species, and therefore, rarity did not affect our extinction selectivity results.

Environment: Similar to redundancy, we found limited differences in occurrences across environments (depths) and some differences in terms of occupancy, where the deepest environment had mostly rare species. Given that our extinction models found an effect mostly in depths between 5-15m and to a lesser degree in depths between 100-200 m (red boxplots) and that both depth categories have abundant and rare species, we consider that abundance did not affect our extinction selectivity results.

Body size: Again there were no differences in occurrences across sizes, but the two smallest sizes had the rarest species in terms of occupancy. These two size categories also showed the highest effect on extinctions. So in the case of size, smallest species are the rarest and more prone to extinction. This is contrary to the expectation of small species being the most abundant. However, it exemplifies the fact that small fossils are less likely to be collected. As such, this find may represent a sampling artifact. Regardless, please note that the actual extinction models account for preservation and sampling biases (see methods) and that the effect of size on extinction was only marginal (see figure 2a).

Based on these results, we argue species abundance/rarity **did not confound the effects of functional traits and redundancy on extinction because extinction was affected by trait categories that had both rare, or abundant species**. Crucially, this is true for the case of redundancy, which is the main factor affected by extinction in our study.

In conclusion, the additional tests we carried out to tackle their question, which are now included in the revised manuscript, show that our results are not the product of biases in trait distribution between rare and abundant species. We think that these tests serve to clarify the robustness of our results and are thankful to the reviewer for raising this important point.

Minor comments:

Figure S1: y-axis labels are missing, and one of the trait categories is missing from three of the panels. The caption repeats the information in Table S2, and I suggest showing trait categories directly on the x-axis as tick labels to enhance readability.

We have modified this figure to reflect the number of occurrences per species per trait. In so doing, we followed the reviewer's suggestions.

Check the order of appearance of Supplementary figure references in the main text.
Checked and fixed.

Figure S6 seems identical to the final panel in Figure 3b. Redundancy -> extinction.
Not applicable anymore.

No need to repeat the numbers in line 262 and line 240.
Done.

Lines 278-9: Please explain how the pattern in Fig S5 can be described as "overall high".
We rephrased this in the revised manuscript clarifying that the preservation rates are within the range or rates (or higher) that have been tested in previous simulation studies. Under these rates we expect the PyRate method to perform well.

Appendix C

Dear Professor Hutchinson,

We are grateful for the invitation to submit another revised version of our paper following its initial review.

Below we provide a point-by-point response (in blue) to the editor and reviewer's comment.

We hope this improved version is now suitable for publication in *Proceedings of the Royal Society B*.

On behalf of all co-authors, thank you very much for considering our manuscript.

Thank you for your consideration,

Christine D. Bacon
Associate Professor

University of Gothenburg, Gothenburg, Sweden

Gothenburg Global Biodiversity Centre, Sweden

Universidad CES, Medellín, Colombia

christinedbacon@gmail.com

Associate Editor

Board Member: 1

Comments to Author:

Bacon and colleagues have done an excellent job responding to the reviewer concerns and queries. I commend the authors for the thoroughness of their revision. I have very few concerns remaining, aside from the issue of rarity mentioned by the Reviewer. The Reviewer suggests a number of analyses that the authors should consider to broaden their results and explore further the relationship between rarity and redundancy-extinction link.

We thank the editor for their comments.

We have performed an additional sensitivity analysis to address the issue of rarity, as suggested by the reviewer. Our results confirm that species occupancy does not affect our results derived from our multi-trait-dependent extinction (MTE) model. We have included these new results in the supplementary material and have provide a detailed response below.

It is worth noting that when doing these new analyses, we found a small issue with our plots. Essentially, when plotting figure 2, we only used ~70% of our data instead of 100%. We have fixed this and as a result, our figure 2 has slightly changed. Although the effect of redundancy in extinction rates remained the same, the effect of body size increased and the effect of environment decreased compared with the previous figure (see below). Notably, the effects of diet, life habitat and locomotion on extinction rates are even smaller than shown before. These changes; however, did not affect our results and conclusions: very high and high redundancy, and very small and small size have the highest effects on extinction rates, whereas shore-face environments have the lowest extinction rates. These results are the same as in the previous version. We edited manuscript accordingly, and marked all changes in blue for easy identification by the editor. These changes, are very minor.

Reviewer(s)' Comments to Author:

Referee: 4

Comments to the Author(s).

Again, I wish to commend the authors on an overall solid and impressive study. I am very pleased to see that the authors have followed up on my suggestion to compute simple species occupancy on a spatial grid, to assess the potential influence of abundance/rarity on their results. Unfortunately, however, the authors' assessment and dismissal of any effect of rarity is not very convincing and makes me more curious about the potential role of rarity.

We thank the reviewer for their comments. We have performed new sensitivity analyses that confirm that rarity, measured as species occupancy, does not affect our results, as described below.

I hasten to add that a link between rarity and redundancy/extinction could be important for understanding ecological mechanisms, it is not simply a matter of statistical artifact or bias. Indeed, the authors concede in their response letter that this is an important issue (although not mentioned anywhere in the main text of their revised manuscript).

The authors address this issue by generating box plots of occupancy for each factor/trait found to have a significant effect on extinction (Figure S8). I suggested simple occupancy as a proxy for abundance due to the near universality of a positive abundance-occupancy relationship. The authors also include analogous box plots using occurrences, but unlike occupancy, raw occurrences are not a valid proxy for abundance. In other words, only the right-hand panels of Figure S8 are relevant for testing the effect of rarity.

In their description of the box-plot graphics (Fig. S8), the authors emphasize that the factor/trait categories found to be selected against (highlighted in red) include not only rare species, but also abundant species. From this observation they conclude that rarity did not affect their extinction selectivity results. In their words (Supplement p. 6): "We found that species abundances did not confound the effects of functional traits and redundancy on extinction because extinction was affected by trait categories that had both rare, or abundant species. Crucially, this is true for the case of redundancy, which is the main factor affected by extinction in our study."

However, what is most striking in the box plots (Fig. S8, right panels only) is that four out of the six extinction-related categories have the lowest observed occupancy values. Out of the 17 categories considered, only four have 75% of species with occupancy of 3 grid cells or less and a median occupancy of a single grid cell. Those four are all among the six that have a significant effect on extinction, including the most prominent category of 'very high redundancy'. Conversely, the categories with the highest occupancy values (as indicated by the 75th percentile top of the IQR box) are all among the categories with the weakest effect on extinction (cf. Fig. 2; including 'very low redundancy', 'shoreface', and 'very large/large'). It is not at all clear how these box plots support the authors' conclusion that abundance/rarity did not affect the extinction selectivity. Given the size of the data set, small but consistent differences in abundance distribution may very well play a role in the extinction selectivity, and the authors should not

leave room for doubt.

In light of the reviewer's comments, we have removed figure S8 and replaced it with a figure that shows the results of our new sensitivity analyses (see below).

With the overwhelming majority of species being rare (low occupancy), visual inspection of highly skewed box plots may not be the most useful tool for assessing if/how rarity affects the results of their analysis. In light of the authors' otherwise sophisticated approach to probabilistic inference, I believe they are in a good position to perform a more rigorous sensitivity analysis on the effect of abundance/rarity. I don't have personal experience with the specific methods used by the authors, so I don't know what would be the best approach, but here are some tentative suggestions:

i) Is it possible to perform a sensitivity analysis on the inclusion/exclusion of rare/abundant species, along the lines of the analysis the authors have performed on the inclusion/exclusion of traits (Fig. S7)? This could be problematic due to the numerical dominance of very rare species.

ii) Would it be possible to standardize the abundance (occupancy) distributions across trait categories by subsampling to ensure each category has the same representation of rare/abundant species? This would require Monte Carlo iteration, but should in principle give an idea of the robustness of the extinction selectivity results when variation in occupancy is neutralized.

iii) Why not include occupancy as a factor in the multi-trait dependent extinction model? Sure, it's a rough proxy, but so are the functional traits. Intuitively, this would be the ecologically most meaningful approach.

It is not my intention to make the authors jump through hoops. I would like to see this work published, and therefore I want to help make sure the conclusions are well supported. A more rigorous assessment/sensitivity test of the effect of abundance/rarity seems well within reach and would significantly strengthen the authors conclusions. As I argued in my first review, even if rarity is found to affect the redundancy-extinction link, the relationship between rarity and redundancy should advance our mechanistic understanding of the ecology of extinction and widen the implications of this study.

We have followed the suggestions provided and have performed additional analyses that combine two of the possibilities suggested by the reviewer. Accordingly, we ran a sensitivity test to assess whether the imbalance between rare and abundant species had an impact on our MTE models.

First, we followed the reviewer's suggestion and calculated occupancy as the number of grid cells (1x1 degree) in which fossils of each species were found.

Second, we categorized species occupancy in three classes: 1) species occupying one cell [low occupancy]; 2) between 2 and 5 cells [medium occupancy]; and 3) more than 5 cells [high occupancy]. As predicted, there was a strong imbalance between occupancy classes, with 2425, 1637, and 358 species in each occupancy class, respectively.

Third, to assess whether this imbalance affected the results of the MTE analyses, we generated 10 additional datasets in which species in the low and intermediate occupancy categories were subsampled to reach the number of high occupancy species (358 species; the lowest number of species in all categories). Thus, the occupancy categories in subsampled datasets were equally represented.

Finally, we analyzed the subsampled datasets under the MTE model to verify if the effect of the three traits found to correlate significantly with extinction based on the full dataset (i.e., redundancy, body size, and environment) were a result of an over-representation of rare species.

As shown in the figure below (figure S8 in the revised manuscript), the estimated effects of redundancy, body size, and environment remained highly consistent with those inferred from the full set of mollusk species. Accordingly, very high and high redundancy, and very small and small body sizes have the highest effect on extinctions; and shore-face environments have the lowest effect on extinction (see main text and figure 2).

Based on this new analysis, we conclude that our MTE results are robust, and that differences in occupancy did not affect the effect that redundancy and traits had on extinction.